# DyPE: Dynamic Position Extrapolation for Ultra High Resolution Diffusion

**Noam Issachar** [*] **Guy Yariv** [*] **Sagie Benaim** **Yossi Adi** **Dani Lischinski** **Raanan Fattal**

The Hebrew University of Jerusalem

## Abstract

Diffusion Transformer models can generate images with remarkable fidelity and detail, yet training them at ultra-high resolutions remains extremely costly due to the self-attention mechanism's quadratic scaling with the number of image tokens. In this paper, we introduce Dynamic Position Extrapolation (DYPE), a novel, training-free method that enables pre-trained diffusion transformers to synthesize images at resolutions far beyond their training data, with no additional sampling cost. DYPE takes advantage of the spectral progression inherent to the diffusion process, where low-frequency structures converge early, while high-frequencies take more steps to resolve. Specifically, DYPE dynamically adjusts the model's positional encoding at each diffusion step, matching their frequency spectrum with the current stage of the generative process. This approach allows us to generate images at resolutions that exceed the training resolution dramatically, *e.g.*, 16 million pixels using FLUX. On multiple benchmarks, DYPE consistently improves performance and achieves state-of-the-art fidelity in ultra-high-resolution image generation, with gains becoming even more pronounced at higher resolutions. Project page is available at https://noamissachar.github.io/DyPE/.

## 1. Introduction

Diffusion Transformers (DiTs) (Peebles & Xie, 2022) have recently emerged as a powerful class of generative models, combining the stable training dynamics of diffusion (Ho et al., 2020; Song et al., 2020) with the expressiveness and

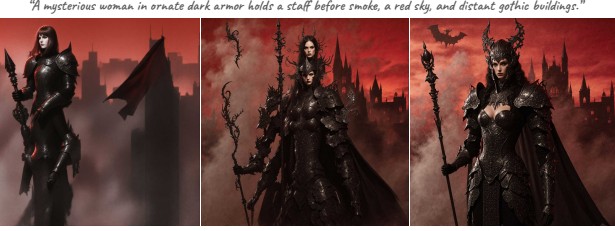

*"A mysterious woman in ornate dark armor holds a staff before smoke, a red sky, and distant gothic buildings."*

FLUX | YaRN | *Dy-YaRN*

*Figure 1.* DYPE enables pre-trained diffusion transformers to generate ultra-high-resolution images (16M+ pixels) without retraining and without inference overhead, solely by coordinating the positional encoding with the diffusion's progression. We compare the baseline FLUX, YaRN, and DYPE, specifically the DY-YaRN variant, both applied on top of FLUX, at $4096 \times 4096$ resolution.

scalability of transformers (Vaswani et al., 2017; Kaplan et al., 2020; Hoffmann et al., 2022). While this architecture fueled progress across large-scale vision (Dosovitskiy et al., 2021), training these models to ultra-high resolutions (*e.g.*, $4096^2$ and beyond) remains a formidable challenge: the quadratic complexity of self-attention in the number of image tokens drives up memory and compute costs, making direct training infeasible.

This limitation is analogous to the *long-context* challenge in large language models (LLMs), where transformers are trained with a fixed context horizon, but are expected to perform on much longer sequences during inference. The positional encoding (PE) mechanism is central to this generalization, as it dictates how transformers align and extrapolate positional relations across unseen ranges. Rotary Positional Embeddings (RoPE) (Su et al., 2021) are widely adopted but degrade when extrapolated beyond the training range. This has motivated inference-time adaptations such as position interpolation (PI) (Chen et al., 2023b), NTK-aware rescaling (Peng et al., 2023), and YaRN (Peng et al., 2023), which adjust the frequency spectrum to better preserve long-range dependencies.

In image generation, these LLM-derived schemes were adapted to accommodate aspect-ratio changes and moderate increases in resolution (Lu et al., 2024; Wang et al., 2024). However, these static approaches do not account for the distinctive *spectral progression* of the diffusion process,

---

[*]Equal contribution . Correspondence to: Noam Issachar <noam.issachar@mail.huji.ac.il>, Guy Yariv <guy.yariv@mail.huji.ac.il>.

*Proceedings of the $43^{rd}$ International Conference on Machine Learning*, Seoul, South Korea. PMLR 306, 2026. Copyright 2026 by the author(s).

where low-frequency structures are generated in the first sampling steps, while high-frequency details are resolved later (Rissanen et al., 2023; Hoogeboom et al., 2023). As shown in Zhuo et al. (2024), aligning with these dynamics can facilitate better resolution extrapolation. These observations naturally lead to our guiding question: *How should positional embeddings be dynamically adjusted to reflect the spectral progression of the diffusion process?*

In this work, we analyze the spectral dynamics of the inverse diffusion process. Specifically, we assess the synthesis timeline at which each frequency component of the generated sample evolves as a function of the sampling step. This analysis shows that low-frequency Fourier components converge to their final values much earlier while high-frequency components evolve throughout the denoising. This fine-grained observation allows us to design our *Dynamic Position Extrapolation (*DyPE*)*, which exploits this progression: as sampling continues, the PE shifts more emphasis from the already-solidified low frequencies to the evolving high-frequency bands. By dynamically tailoring the PE's spectral allocation, DyPE better serves the instantaneous needs of the diffusion operator throughout its sampling course.

This *training-free* strategy greatly improves generalization, allowing a pre-trained FLUX model (Labs, 2024) to generate images at ultra-high resolutions (exceeding 16M pixels), as shown in Fig. 1. We evaluate DyPE using quantitative metrics for image quality and prompt fidelity, alongside qualitative and human evaluations. The results show that DyPE achieves consistent improvements in ultra-high-resolution synthesis across multiple benchmarks and resolutions, all without retraining or additional sampling costs.

## 2. Preliminaries

### 2.1. Diffusion Models

Diffusion models progressively evolve samples from a latent pure-noise, Gaussian distribution $\mathcal{N}(0, I)$, towards a target distribution $q(x)$ via a sequence of intermediate mixture distributions. The process is governed by a time parameter $t \in [0, 1]$ that defines the mixture variables $x_t$, by:

$$x_t = \alpha_t x + \sigma_t \epsilon, \qquad x \sim q(x), \;\; \epsilon \sim \mathcal{N}(0, I), \quad (1)$$

where the schedule coefficients $\alpha_t$ and $\sigma_t$ are chosen to achieve the endpoints $x_0 = x$ (pure data) and $x_1 = \epsilon$ (pure Gaussian noise). We denote these mixture distributions by $q_t$.

Different schedules $\alpha_t$ and $\sigma_t$ correspond to different formulations, e.g., Variance Preserving (Ho et al., 2020; Song et al., 2020) and Flow Matching (Lipman et al., 2022; Liu et al., 2022). The latter using the linear schedule $\alpha_t = 1 - t$ and $\sigma_t = t$, which we adopt in our derivation.

### 2.2. Rotary Positional Embeddings and Position Extrapolation

**Positional Embedding (PE).** The transformer block, which is the basis of DiT, is permutation equivariant. Thus, a positional encoding mechanism is necessary to properly model the strong spatial dependencies in natural images (Le-Cun & Bengio, 1998). Early solutions use fixed sinusoidal positional embedding (Vaswani et al., 2017; Dosovitskiy et al., 2021), learned absolute embeddings (Devlin et al., 2019; Radford et al., 2019), or relative positional embeddings (Press et al., 2021). More recently, the *Rotary Positional Embeddings* (RoPE) (Su et al., 2021) emerged as a more effective alternative which provides the relative positions in the query–key interactions.

More specifically, RoPE represents a position coordinate $m$ as a set of 2D rotations at different frequencies. The number of frequencies is determined and *limited* by $D = d_{\mathrm{model}}/2$, where $d_{\mathrm{model}}$ is the hidden model dimension. The frequencies $\theta_d$ are typically obtained from a geometric series,

$$\theta_d = \theta_{\mathrm{base}}^{\frac{d}{D-1}}, \qquad d = 0, \ldots, D - 1, \qquad (2)$$

with corresponding wavelength $\lambda_d = 2\pi/\theta_d$, where $\theta_{\mathrm{base}}$ is a model hyper-parameter. We note that in case of 2D images RoPE is applied *axially*: half of the hidden vector is rotated horizontally, and the other half vertically. Thus this axial decomposition enables RoPE to encode relative offsets along each axis independently, considering the spatial structure of images (Heo et al., 2024).

As discussed above, training DiT models at high resolutions incurs substantial memory and compute cost. Applying a model at higher resolutions than it was trained on, suffers from degraded performance as illustrated in Fig. 1. This shortcoming spurred the development of inference-time positional encoding adaptations for a better generalization. Before we survey these approaches, let us establish useful notations from Peng et al. (2023).

Assuming the training context length, is $L$, and $L'$ is the extended context, we define the *scaling factor s* by:

$$s = L'/L. \qquad (3)$$

Moreover, the different extrapolation methods can be characterized by their action over the spatial coordinate $m$ and frequencies $\theta_d$ that they represent, namely:

$$m \mapsto g(m), \qquad \theta_d \mapsto h(\theta_d), \qquad (4)$$

where $g$ and $h$ are method-specific transformations.

**Position Interpolation (PI)** is an early approach (Chen et al., 2023b), that rescales uniformly the position $m$ to the new context length $L'$ by:

$$g(m) = m/s, \qquad h(\theta_d) = \theta_d. \qquad (5)$$

This mapping resamples the waves $\cos(m\theta_d), \sin(m\theta_d)$ at a finer rate in the larger context grid $L'$, and while it correctly reproduces the lower end of the spectrum, it fails to reach the new grid's higher frequency band. While large scale content is properly synthesized in this approach, the missing high-frequencies manifest as blurriness and lack of fine detail, as discussed in Appendix A.

**NTK-Aware Interpolation.** To address this problem, the *Neural Tangent Kernel (NTK-aware)* interpolation (Peng et al., 2023) applies different scaling to the low and high frequencies, by:

$$g(m) = m, \qquad h(\theta_d) = \frac{\theta_d}{s^{2d/(D-2)}}. \qquad (6)$$

Thus, the low frequencies (large $\lambda_d$) remain nearly unchanged in the new grid as in PI, by trading off the representation of the high frequencies (small $\lambda_d$) due to the compression resulting from accommodating the higher band of the larger context $L'$.

**YaRN.** Yet another RoPE extensioN, or *YaRN* (Peng et al., 2023) extends the latter in two ways. The first is the *NTK-by-parts* interpolation, which splits the spectrum to three bands, where different mappings are applied, namely:

$$g(m) = m, \qquad h(\theta_d) = (1 - \gamma(r(d))) \frac{\theta_d}{s} + \gamma(r(d)) \theta_d, \qquad (7)$$

where $r(d) = L/\lambda_d$. The ramp $\gamma(r)$ provides a smooth transition from PI stretching to no scaling:

$$\gamma(r) = \begin{cases} 0, & r < \alpha, \\ \frac{r-\alpha}{\beta-\alpha}, & \alpha \le r \le \beta, \\ 1, & r > \beta, \end{cases} \qquad (8)$$

where $\alpha, \beta$ set the bands' boundaries. Also here the bands are scaled non-uniformly, with more flexibility to control the allocation trade-offs made by NTK-aware interpolation.

The second extension is the *attention scaling*, where attention logits are modified by a factor $\tau(s) = 0.1 \ln(s) + 1$. The resulting attention mechanism is defined as

$$\text{Attn}_{\text{YaRN}}(q_i, k_j) = \text{softmax}\left( \tau(s) \cdot \frac{q_i^\top k_j}{\sqrt{d_{\text{model}}}} \right). \qquad (9)$$

This allows to counterbalance (reduce) the increase in entropy of the attention weights due to the introduction of additional keys in the larger context $L'$.

## 3. Method

We now present DYPE. We first analyze the spectral dynamics of the diffusion process, showing how different frequency modes evolve over time (Sec. 3.1). Based on this analysis, we derive DYPE, which dynamically adjusts positional encoding to match these dynamics (Sec. 3.2).

### 3.1. Evolution of Frequency Modes in the Diffusion Process

The simple mixture formulation in Eq. 1 allows us to derive a complementary perspective in Fourier space, as given by:

$$\hat{x}_t = (1 - t)\hat{x} + t\hat{\epsilon}, \qquad (10)$$

where $\hat{(\cdot)}$ denotes the Fourier transformed signals. The i.i.d noise vectors $\epsilon$ have a white (constant) Power Spectrum Density (PSD), and the data of natural images, $x_t$, is known to have a well-characterized PSD with a power-law decay of $\propto 1/f^\omega$ where $\omega \approx 2$ (van der Schaaf & van Hateren, 1996; Hyvrinen et al., 2009), as function of frequency $f$. These terms allow us to explicitly describe the time-dependent mean PSD in Eq. 10, given by

$$\overline{\|\hat{x}_t\|^2}_f = (1 - t)^2 C/f^\omega + t^2, \qquad (11)$$

which results from computing the mean PSD of $x_t$, denoted by $\overline{(\cdot)}$, according to Eq. 10, and noting that the covariance $\langle \hat{x}, \hat{\epsilon} \rangle = 0$ due to independence. The constant $C$ is a characteristic PSD scale of the particular data distribution. Fig. 2a depicts the empirical evaluation of the averaged PSD computed over samples generated by a denoiser trained on ImageNet (Russakovsky et al., 2015). The function reveals the smooth transition between the two spectra and reflects the growth of low-frequency image structures and the decay of noise alongside the emergence of high-frequency fine-details, as predicted by Eq. 11.

The question we would like to address here is whether this evolution is fully "active" during the entire sampling process and at all the frequencies, or whether it shows some regularities which we can exploit for a better allocation of the represented spectrum.

To assess the rate at which these modes evolve, we consider a *progression map* relating each frequency component $f$ to a progress index, $0 \le \gamma(f, t) \le 1$, that indicates the relation of its log-PSD value at time $t$, *i.e.*, $\log(\|\hat{x}_t\|^2_f)$, in relation to its endpoints. By utilizing the fact that the transition described by Eq. 11 is monotonic, this index is easily obtainable by

$$\gamma(f, t) = \frac{s(t)_f - s(0)_f}{s(1)_f - s(0)_f} \qquad (12)$$

where $s(t)_f = \log(\|\hat{x}_t\|^2_f)$.

Fig. 2b shows this progression map where a clear observation can be made. While the higher frequency components show a fairly constant evolution throughout the sampling process, the lower frequencies appear to evolve faster, and more importantly, *cease* to evolve fairly early in sampling. Assuming that the evolving modes depend more on their corresponding frequency representation in the PE than the

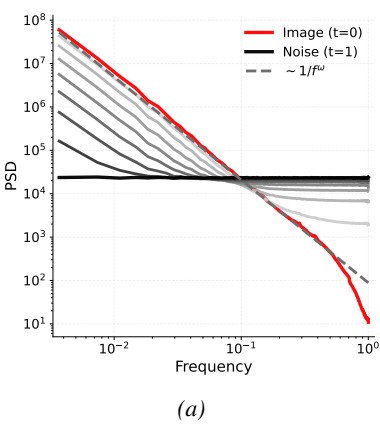

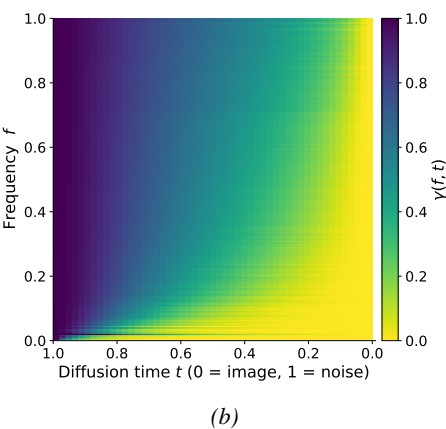

*(a)*                                     *(b)*

*Figure 2.* **Spectral Evolution of Samples in the Diffusion Process. (a)** shows the average PSD of images produced by a diffusion model, as a function of time $t$. The flat spectrum at $t = 1$ corresponds to the initial Gaussian samples, and the characteristic natural images power-law appears as the process ends ($t = 0$). The combinations of these spectra, corresponding to the mixture distributions $q_t$, are seen at the intermediate steps (gray plots). The *progression map* $\gamma(f, t)$ from Eq. 12 is shown in **(b)**, and measures in relative terms how each Fourier component evolves from pure noise ($t = 1$) to its clean image value ($t = 0$). As seen in the top rows of this map ($t \approx 1$), the high-frequency modes evolve gradually and nearly linearly across the entire reverse process. By contrast, the low-frequency modes converge much faster and cease to change early on, as indicated by the map's saturation (yellow) in the lower rows ($t \approx 0$).

converged ones, the following frequency allocation strategy can be derived: at the beginning of the process, all modes evolve and hence all modes in the finer grid should be accommodated in the PE, *e.g.*, using an existing extrapolation encoding strategies such as YaRN. As the sampling progresses, more and more modes in the lower end of the spectrum convergence and the PE emphasis should be allocated in favor of representing the yet-unresolved higher frequencies.

We further note that frequency extrapolation formulae allocate more low frequency components at the cost of removing higher ones, *e.g.*, in NTK-aware and YaRN. Thus, switching off the extrapolation, as we suggest, has two benefits: (i) more high-frequency modes are represented in the PE, and (ii) the pretrained denoiser will operate in the conditions, namely the PE, it was trained with. These observations serve as a basis for the design of our new approach, DyPE, which we describe next.

This topic is briefly touched by (Zhuo et al., 2024) as part of deriving a new DiT architecture. However, the opposite conclusions are drawn. We discuss this strategy in Appendix B.

### 3.2. Dynamic Position Extrapolation (DyPE)

Our new approach, DyPE, is motivated by two complementary insights. First, as discussed above, the reverse diffusion trajectory exhibits a clear spectral ordering: low-frequency, large-scale structures converge early, while high frequency bands are resolved throughout the sampling process. Second, while the existing positional extrapolation strategies, NTK-aware, and YaRN, are capable of representing the spectrum of the larger context using the limited number of

available modes, $D$, they involve representation trade-offs due to the compression they must employ. Thus, rather than pinpointing both ends of the spectrum at all times and accommodating these trade-offs, our method, DyPE, accounts for the spectral progression and gradually lowers their use to minimize the compression they involve.

We implement this strategy by introducing explicit time dependence into the formulae of PI, NTK-aware, and YaRN. A unifying observation is that all three methods effectively "shut-down" when the scaling factor $s = 1$, *i.e.*, no change in context length. Specifically, in PI, we get $g(m) = m/s = m$; in NTK-aware, $h(\theta_d) = \theta_d s^{2d/(D-2)} = \theta_d$; and YaRN, which combines the components of both PI and NTK-aware, likewise collapses to no scaling.

Consequently, we define the following family of time-parameterized scalings,

$$\kappa(t) = \lambda_s \cdot t^{\lambda_t}, \tag{13}$$

with tunable hyperparameters $\lambda_s$ and $\lambda_t$. Early in sampling ($t \approx 1$), this formula yields near-maximal scaling $\kappa(1) = \lambda_s$; late in sampling ($t \approx 0$), it approaches no scaling $\kappa(0) = 1$.

The exponent $\lambda_t$ controls how scaling attenuates over time, allowing us to align the evolution of frequency emphasis with diffusion's progression. The multiplier $\lambda_s$ sets the maximal scaling that DyPE attains; in principle it reflects the ratio between the desired and the training context lengths.

Finally, let us now go through the resulting extrapolation strategies from plugging $\kappa(t)$ into these methods, either by replacing the fixed scaling parameters $s$, or controlling the thresholds in YaRN.

**DY-PI.** PI in Eq. 5 uses uniform position scaling. We make it step-aware by exponentiating the scale factor by $\kappa(t)$:

$$g(m,t) \;=\; \frac{m}{s^{\kappa(t)}}, \qquad h(\theta_d, t) \;=\; \theta_d. \qquad (14)$$

Early sampling steps ($t \approx 1$) apply stronger compression to stabilize structure, while later steps ($t \approx 0$) resolve finer detail.

**DY-NTK.** NTK-aware interpolation in Eq. 6 rescales frequencies non-uniformly. Our time-aware variant generalizes this by multiplying the exponent with $\kappa(t)$:

$$g(m,t) \;=\; m, \qquad h(\theta_d, t) \;=\; \frac{\theta_d}{s^{\kappa(t)\cdot 2d/(D-2)}}. \qquad (15)$$

In this scheme. the low frequencies are well-represented at the initial steps, at the cost of compressing the high-frequency band. As the sampling progresses, the low-frequency modes converge, and the higher frequency band representation expands. An illustration of this approach is provided in Appendix A.

**DY-YaRN.** YaRN in Sec. 2.2 combines NTK-by-parts frequency scaling (Eq. 7) with global attention scaling (Eq. 9). Unlike the two methods above, here we introduce time-dependence via $\kappa(t)$ which dynamically adjusts the fixed ramp thresholds $\alpha$ and $\beta$ in Eq. 8, resulting in

$$\gamma(r,t) = \begin{cases} 0, & r < \alpha \cdot \kappa(t), \\ \frac{r - \alpha\cdot\kappa(t)}{\beta\cdot\kappa(t) - \alpha\cdot\kappa(t)}, & \alpha \cdot \kappa(t) \le r \le \beta \cdot \kappa(t), \\ 1, & r > \beta \cdot \kappa(t), \end{cases} \quad (16)$$

and since $\kappa(t)$ is already multiplied by $\alpha$ and $\beta$, we set $\lambda_s = 1$, and hence $\kappa(t)$ in this case reduces to

$$\kappa(t) \;=\; t^{\lambda_t}. \qquad (17)$$

Being a monotonic increasing function, the scheduler $\kappa(t)$ dynamically shifts the ramp boundaries towards 1, *i.e.*, no scaling, as function of the sampling step $t$, which meets our design goal.

# 4. Experiments

We evaluate the effectiveness of DYPE across multiple aspects of high-resolution image generation, covering both global structure (low-frequency aspects such as text–image alignment) and fine detail (high-frequency aspects such as texture fidelity).

We first apply DYPE on top of FLUX (Labs, 2024), with evaluations on two established benchmarks, Draw-Bench (Saharia et al., 2022) and Aesthetic-4K (Zhang et al., 2025a), including automatic metrics, human evaluation, and resolution-scaling analysis (Sec. 4.1). We then extend evaluation to class-conditional image synthesis on FiTv2 (Wang

*Figure 3.* Zoom-in comparison at $4096^2$ of DY-YaRN vs. YaRN. Additional example can be found in Fig. 17 in the Appendix.

et al., 2024) (Sec. 4.2). We also include zoom-in studies to highlight improvements in preserving high-frequency details (Fig. 3). Furthermore, in Appendix D, we present an ablation study examining design choices, focusing on (i) scheduler variants for DY-NTK-aware and (ii) timestep incorporation strategies for DY-YaRN. Finally, additional results are provided in Appendix E, covering additional DiT-based architectures (Qwen-Image (Wu et al., 2025)), high-resolution video generation (Wan et al., 2025), and high-resolution image editing tasks, panorama generation, and more visual examples. Implementation details are provided in Appendix C.

## 4.1. Ultra-High-Resolution Text-to-Image Generation

In Sec. 4.1.1 we evaluate DYPE against Position-Extrapolation-based approaches and in Sec. 4.1.2 we evaluate DYPE against more general baselines as custom in previous works (Bu et al., 2025).

### 4.1.1. COMPARISON WITH POSITION-EXTRAPOLATION BASELINES

We evaluate DYPE on top of the pre-trained FLUX (Labs, 2024), specifically the FLUX.1-Krea-dev version, whose effective generation resolution is $1024 \times 1024$. As primary baselines, we use FLUX itself and, in test time only, apply on top of FLUX the positional–embedding extrapolation methods NTK-aware and YaRN, adapted to vision by applying them independently on the $x$ and $y$ axes. We also compare with Time-Aware Scaled RoPE (TASR) (Zhuo et al., 2024), which interpolates from PI to NTK-aware scaling as denoising advances (discussed in Appendix B). On top of these, we evaluate our DYPE, including both DY-NTK-aware and DY-YaRN.

**Benchmarks.** As for benchmarks, we first consider Draw-Bench (Saharia et al., 2022), a set of 200 text prompts for evaluating text-to-image models across multiple criteria. Following Ma et al. (2025); Chachy et al. (2025), we measure: (i) text-image alignment using CLIP-Score (Hessel et al., 2022), a similarity metric between image and text embeddings based on CLIP (Radford et al., 2021), (ii) human preference alignment using ImageReward (Xu et al., 2023),

*Table 1.* High-resolution image generation on DrawBench and Aesthetic-4K on multiple resolutions. Each row reports CLIPScore (CLIP), ImageReward (IR), Aesthetics (Aesth) for DrawBench, and CLIP, IR, Aesth, and FID for Aesthetic-4K. All methods are built on FLUX.

| Method | $2048 \times 3072$ DrawBench CLIP↑ | IR↑ | Aesth↑ | Aesthetic-4K CLIP↑ | IR↑ | Aesth↑ | FID↓ | $3072 \times 2048$ DrawBench CLIP↑ | IR↑ | Aesth↑ | Aesthetic-4K CLIP↑ | IR↑ | Aesth↑ | FID↓ |
|---|---|---|---|---|---|---|---|---|---|---|---|---|---|---|
| FLUX | 26.64 | -0.28 | 5.14 | 28.64 | 0.32 | 6.11 | 186.31 | 26.56 | 0.16 | 5.33 | 28.74 | 0.97 | 6.17 | 148.29 |
| NTK | 27.68 | 0.21 | 5.31 | 29.13 | 0.99 | 6.49 | 180.87 | 27.28 | 0.51 | 5.39 | 28.97 | 1.17 | 6.25 | 146.74 |
| TASR | 27.86 | 0.30 | 5.15 | 29.13 | 0.97 | 6.12 | 201.40 | 27.40 | 0.55 | 5.22 | 29.05 | 1.15 | 5.95 | 186.21 |
| DY-NTK | **27.91** | **0.48** | **5.54** | **29.14** | **1.10** | **6.56** | **176.13** | **27.44** | **0.60** | **5.55** | **29.11** | **1.21** | **6.53** | **146.40** |
| YaRN | 28.27 | 0.52 | 5.63 | 29.28 | 1.01 | 6.59 | 179.54 | 27.79 | 0.62 | 5.48 | 29.12 | 1.24 | 6.49 | 147.12 |
| DY-YaRN | **28.43** | **0.71** | **5.69** | **29.44** | **1.17** | **6.61** | **179.51** | **28.17** | **0.81** | **5.68** | **29.20** | **1.28** | **6.51** | **146.84** |

| Method | $3072 \times 3072$ DrawBench CLIP↑ | IR↑ | Aesth↑ | Aesthetic-4K CLIP↑ | IR↑ | Aesth↑ | FID↓ | $4096 \times 4096$ DrawBench CLIP↑ | IR↑ | Aesth↑ | Aesthetic-4K CLIP↑ | IR↑ | Aesth↑ | FID↓ |
|---|---|---|---|---|---|---|---|---|---|---|---|---|---|---|
| FLUX | 25.11 | -0.53 | 5.01 | 28.62 | 0.46 | 6.16 | 187.96 | 16.43 | -1.97 | 3.29 | 25.50 | -0.73 | 5.42 | 195.68 |
| NTK | 26.07 | -0.14 | 5.05 | 28.68 | 0.96 | 6.45 | 182.38 | 17.49 | -1.88 | 3.57 | 24.88 | -0.54 | 5.50 | 203.85 |
| TASR | 26.87 | 0.18 | 5.01 | 28.79 | 1.00 | 6.01 | 194.23 | 21.21 | -1.69 | 3.56 | 25.09 | -0.09 | 5.96 | 221.39 |
| DY-NTK | **27.02** | **0.30** | **5.36** | **28.83** | **1.10** | **6.57** | **179.98** | **21.51** | **-1.22** | **4.25** | **28.06** | **0.79** | **6.42** | **183.72** |
| YaRN | 27.92 | 0.41 | 5.37 | 29.26 | 1.14 | 6.67 | 184.16 | 25.71 | -0.34 | 4.85 | 28.57 | 0.85 | 6.47 | 192.19 |
| DY-YaRN | **28.12** | **0.66** | **5.55** | **29.75** | **1.24** | **6.70** | **179.82** | **26.94** | **0.15** | **5.17** | **29.28** | **1.09** | **6.67** | **186.00** |

*"A woman with short hair and a black dress stands in a forest, holding an owl with large, outstretched wings..."*

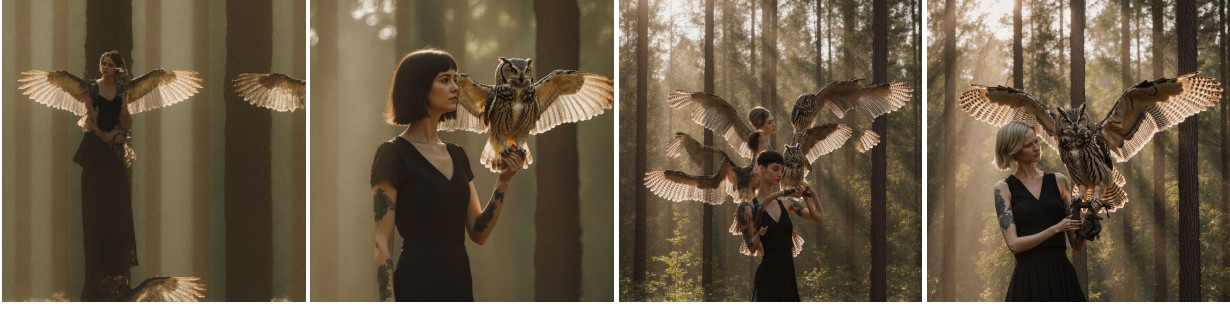

*"A decorative vase with floral branches and white blossoms sits on a light cloth, accompanied by a shiny red apple."*

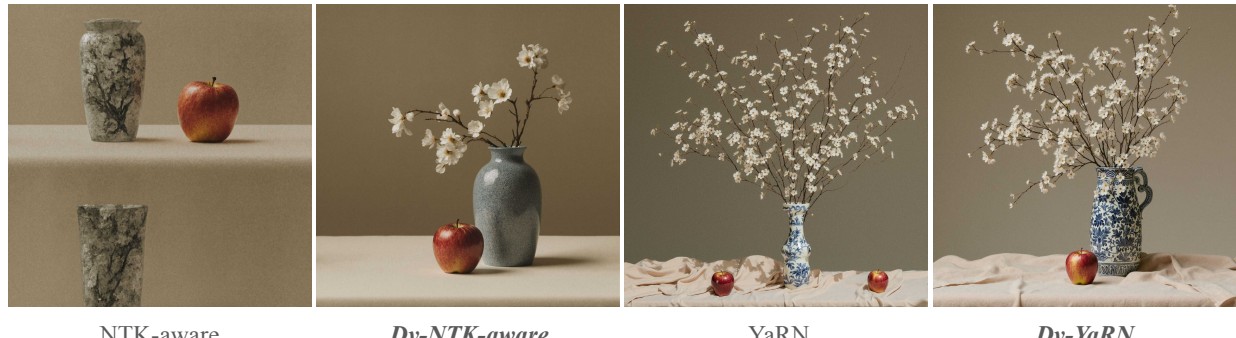

| NTK-aware | *Dy-NTK-aware* | YaRN | *Dy-YaRN* |
|---|---|---|---|

*Figure 4.* Qualitative results at $4096^2$ resolution using two representative prompts from Aesthetic-4K.

a reward model trained on large-scale human feedback for generated images, and (iii) image aesthetics using Aesthetic-Score-Predictor (Schuhmann et al., 2022), a model trained to predict human aesthetic judgments. Additionally, to specifically assess fine-grained, ultra-high-resolution fidelity, we evaluate on Aesthetic-4K (Zhang et al., 2025a). We use its

4K subset (Aesthetic-Eval@4096), which comprises 195 curated image–prompt pairs, and downsample them to match the target test resolutions for fair comparison. Following the official protocol, we report (i) CLIPScore, (ii) ImageReward, (iii) Aesthetics score, and (iv) FID (Heusel et al., 2017), which assesses the fidelity and diversity of generated

images based on the distributional distance between real and generated features.

**Results.** Quantitative results across different resolutions and aspect ratios are presented in Tab. 1, with Fig. 4 showing side-by-side comparisons on Aesthetic-4K. Additional qualitative results are provided in the Appendix for Draw-Bench (Fig. 15) and Aesthetic-4K (Fig. 16). As can be seen in Fig. 1, FLUX exhibits repeating artifacts at ultra-high resolutions, revealing the periodicity of the sinusoidal positional encoding when extrapolated to larger spatial contexts, as further illustrated in Appendix A. We also observe that FLUX performs relatively better on landscape resolutions than portrait, likely reflecting a training-set bias. Notably, once DYPE is applied, this gap widens in favor of our approach on portrait settings as well, indicating that DYPE helps mitigate this limitation. Importantly, the advantage of DYPE becomes increasingly pronounced as the generation resolution grows (e.g., up to $3072^2$ and $4096^2$), underscoring the effectiveness of our method for ultra-high-resolution synthesis in diffusion transformers. Further visual results are presented in the Appendix.

**Perceptual Evaluation.** To complement the automatic metrics, we conduct a human study on a curated subset

*Table 2.* Human evaluation on Aesthetic-4K. Each cell reports the percentage of pairwise comparisons in which DYPE was preferred.

| Comp. | Txt↑ | Str↑ | Det↑ |
|---|---|---|---|
| NTK vs. DY-NTK | 88.5 | 88.7 | 88.3 |
| TASR vs. DY-NTK | 70.5 | 80.6 | 94.2 |
| YaRN vs. DY-YaRN | 90.1 | 87.3 | 88.1 |

of 20 prompts from Aesthetic-4K, obtained by sampling every fourth entry to ensure uniform coverage. We consider 50 raters and present them with pairwise comparisons at $4096^2$ resolution, generated on FLUX. Each prompt yields three comparisons: (i) NTK-aware vs. DY-NTK-aware, (ii) Time-Aware Scaled RoPE (TASR) vs. DY-NTK-aware, and (iii) YaRN vs. DY-YaRN. For each pair, participants answer the following three questions: (i) *Which image is more aligned with the given text prompt?* (ii) *Which image has better overall geometry and structure (coherent shapes, correct proportions, fewer distortions)* and (iii) *Which image has more aesthetic and realistic textures and fine details?* Results, summarized in Tab. 2 shows that DYPE consistently achieves superior quality, with preference rates ranging from about 70.5% to 94.2%.

**Resolution Scaling Analysis.** We next investigate the resolution limit beyond which methods fail. Using 20 Aesthetic-4K prompts sampled at intervals of 10, we evaluate FLUX,

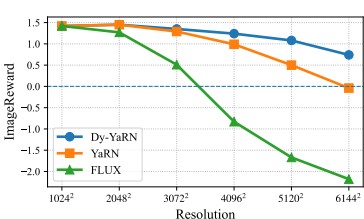

*Figure 5.* Scaling analysis.

YaRN, and our DY-YaRN across six square resolutions from $1024^2$ to $6144^2$, reporting ImageReward (Fig. 5). The trend shows FLUX degrades sharply at $3072^2$ and YaRN at $4096^2$, while our method remains stable across scales until experiencing degradation at $6144^2$.

### 4.1.2. COMPARISON WITH OTHER APPROACHES — GENERAL BASELINES

In addition to PE approaches, we also compare with a broad set of non-PE methods spanning three families of high-resolution diffusion pipelines:

(i) *DiT-based methods* - UltraPixel (Ren et al., 2024) (which relies on SD3 (Esser et al., 2024)), Diffusion-4K (Zhang et al., 2025a) (which requires model fine-tuning), HiFlow (Bu et al., 2025), and I-Max (Du et al., 2024b) (both of which rely on multi-stage or progressive upscaling).

(ii) *U-Net–based methods* - DemoFusion (Du et al., 2024a), FreCaS (Zhang et al., 2025b), DiffuseHigh (Kim et al., 2025), and FreeScale (Qiu et al., 2025).

(iii) *Diffusion + Super-Resolution* - FLUX combined with BSRGAN (Zhang et al., 2021).

**Benchmarks.** Consistent with previous works (Bu et al., 2025), our evaluation focuses on *patch-based* fidelity and detail preservation. Specifically, we report: (i) FID (Heusel et al., 2017), (ii) patch-FID, (iii) Inception Score (IS) (Salimans et al., 2016), and (iv) patch-Inception Score (patch-IS). These complementary metrics isolate local structure and texture quality at ultra-high resolutions, allowing us to more precisely assess how well each method maintains fine-grained detail.

**Results.** We evaluate at resolutions $2048^2$ and $4096^2$ to specifically assess local detail preservation at extreme scales. As can be seen in Tab. 3, at $2048^2$ our DY-YaRN variant achieves the best performance across *all* four metrics among all compared methods. At $4096^2$, DY-YaRN attains the best patch-FID and patch-IS, while the DY-YaRN+HiFlow combination yields the best global FID and IS. Notably, for every metric at both resolutions, at least one DYPE-based variant (either DY-YaRN or DY-YaRN+HiFlow) outperforms the baselines, highlighting the effectiveness of our approach. Additionally, in the Appendix, Fig. 14, we include qualitative comparison of our DY-YaRN variant with representative baselines.

### 4.2. Higher-Resolution Class-to-Image Generation

After validating our method on text-to-image generation, we next test whether its consistency gains transfer to the core task of class-conditional generation on ImageNet (Russakovsky et al., 2015). We apply DYPE on FiTv2 (Wang

*Table 3.* Evaluation at $2048^2$ and $4096^2$ resolutions on non-PE approaches. Baselines include U-Net-based methods (*) , tuning-based approaches (§), progressive refinement pipelines (‡), and diffusion combined with super-resolution (†).

| Method | 2048 × 2048 | | | | 4096 × 4096 | | | |
|---|---|---|---|---|---|---|---|---|
| | FID↓ | patch-FID↓ | IS↑ | patch-IS↑ | FID↓ | patch-FID↓ | IS↑ | patch-IS↑ |
| DemoFusion* | 205.59 | 199.00 | 10.03 | 10.49 | 205.86 | 195.69 | 10.93 | 7.92 |
| FreCaS* | 201.07 | 195.73 | 11.45 | 10.52 | 200.95 | 202.41 | 11.14 | 8.02 |
| DiffuseHigh* | 178.08 | 117.43 | 14.47 | 10.68 | 186.25 | 96.99 | 12.62 | 7.56 |
| FreeScale* | 199.87 | 126.45 | 9.89 | 10.55 | 259.24 | 191.23 | 10.66 | 7.74 |
| I-Max‡ | 174.35 | 107.81 | 13.91 | 9.77 | 187.29 | 87.71 | 13.44 | 5.78 |
| FLUX+BSRGAN† | 175.26 | 106.88 | 13.84 | 10.51 | 201.12 | 98.51 | 10.11 | 8.34 |
| UltraPixel‡ | 181.06 | 114.69 | 14.21 | 11.08 | 186.75 | 88.99 | 13.83 | 8.54 |
| Diffusion-4K§ | 178.25 | 98.35 | 13.41 | 10.37 | 198.16 | 94.82 | 13.82 | 4.72 |
| HiFlow‡ | 173.00 | 106.65 | 13.36 | 10.32 | 174.39 | 78.38 | 13.38 | 6.67 |
| **DУ-YaRN (Ours) + HiFlow‡** | 166.71 | 103.06 | 13.60 | 10.74 | **169.46** | 79.64 | **14.18** | 7.06 |
| **DУ-YaRN (Ours)** | **142.74** | **96.34** | **14.76** | **11.95** | 186.00 | **78.33** | 13.96 | **10.13** |

*Table 4.* ImageNet results on FiTv2-XL/2 comparing PI, NTK, TASR, YaRN and our DУPE variants. We report FID↓, sFID↓, Inception Score (IS)↑, Precision↑, and Recall↑ at $320^2$ and $384^2$.

| | FID↓ | | sFID↓ | | IS↑ | | Precision↑ | | Recall↑ | |
|---|---|---|---|---|---|---|---|---|---|---|
| | $320^2$ | $384^2$ | $320^2$ | $384^2$ | $320^2$ | $384^2$ | $320^2$ | $384^2$ | $320^2$ | $384^2$ |
| FiTv2 | 5.79 | 38.90 | 13.7 | 49.51 | 233.03 | 99.28 | 0.75 | 0.39 | 0.55 | **0.57** |
| PI | 11.47 | 118.60 | 21.13 | 85.98 | 197.04 | 23.10 | 0.67 | 0.16 | 0.51 | 0.38 |
| Dу-PI | 7.16 | 39.56 | 17.40 | 51.90 | 231.70 | 99.97 | 0.67 | 0.36 | 0.53 | 0.49 |
| TASR | 10.47 | 74.87 | 15.67 | 66.12 | 222.40 | 101.10 | 0.69 | 0.21 | 0.51 | 0.39 |
| NTK | 6.04 | 36.75 | 14.35 | 47.82 | 232.91 | 104.73 | 0.75 | 0.40 | 0.55 | 0.56 |
| Dу-NTK | 5.22 | 36.04 | **14.29** | 47.46 | 233.11 | 106.45 | 0.75 | 0.42 | **0.57** | 0.56 |
| YaRN | 5.87 | 22.63 | 15.38 | 36.09 | 250.66 | 156.34 | **0.77** | 0.48 | 0.52 | 0.50 |
| Dу-YaRN | **5.03** | **21.75** | 14.48 | **33.92** | **251.73** | **158.02** | **0.77** | **0.49** | 0.53 | 0.52 |

et al., 2024), a flexible DiT trained on multiple resolutions. Specifically, we use the FiTv2-XL/2 variant (675M parameters), which was trained at a maximum resolution of $256 \times 256$, and test it on resolutions $320 \times 320$ and $384 \times 384$. We compare the standard extrapolation methods (PI, NTK-aware, YaRN) and TASR against our DУPE variants (DУ-PI, DУ-NTK, DУ-YaRN). All models are evaluated on the ImageNet validation set ($50,000$ images). We report FID (Heusel et al., 2017), sFID (Nash et al., 2021), Inception Score (IS) (Salimans et al., 2016), Precision, and Recall (Kynkäänniemi et al., 2019). Quantitative results are reported in Tab. 4, show that, as with FLUX, DУPE consistently improves over all vanilla baselines, with DУ-YaRN achieving the best overall performance. Notably, PI severely underperforms relative to base FiTv2, highlighting its ineffectiveness for image generation due to the loss of high-frequency details.

## 5. Related Work

**Diffusion Transformers.** DiT (Peebles & Xie, 2022) have recently emerged as the leading architecture for diffusion-based text-to-image generation (Ho et al., 2020; Song et al., 2020). While U-Nets (Ronneberger et al., 2015) underpinned earlier advances (Rombach et al., 2022; Podell et al.,

2023; Ramesh et al., 2022), DiTs instead adopt transformer-based backbones that naturally capture global context and scale effectively with model and data size, enabling increasingly capable text-to-image models such as FLUX (Labs, 2024), Stable-Diffusion-3 (Esser et al., 2024) and subsequent advances (Gao et al., 2024; Liu et al., 2024a; Chen et al., 2023a). Yet, training these architectures on ultra-high resolutions (*e.g.*, 4K and beyond) remains an open challenge due to the quadratic cost of self-attention, which quickly becomes prohibitive in both memory and computation at such resolutions.

**Ultra-High Resolution Image Synthesis.** Despite this limitation, many works explored *fine-tuning* diffusion models on higher-resolution (Liu et al., 2025; Cheng et al., 2025; Hoogeboom et al., 2023; Liu et al., 2024b; Ren et al., 2024; Teng et al., 2023; Zheng et al., 2024; Zhang et al., 2025a; Huang et al., 2024), yet these remain limited in their ability to scale to ultra-high resolutions due to the expensive tuning phase. Alternatively, patch-based methods (Bar-Tal et al., 2023; Du et al., 2024a; He et al., 2023) aim to reduce costs by *stitching generated regions*, yet often suffer from duplication and local repetition. Input-level techniques suppress undesired semantics (Lin et al., 2024b; Liu et al., 2024c), but are limited to small artifacts and risk information leakage. Complementary strategies improve high-resolution fidelity within U-Net architectures by modifying internal feature processing, such as FreeU (Si et al., 2024), which enhances skip-connection feature mixing, and FAM-Diffusion (Yang et al., 2025), which introduces frequency modulation for sharper high-resolution outputs. More recently, *tuning-free* methods that synthesize full images without retraining (Qiu et al., 2025; Cao et al., 2025; Haji-Ali et al., 2024; Hwang et al., 2024; Jin et al., 2023; Kim et al., 2025; Lee et al., 2023; Lin et al., 2024a; Zhang et al., 2024) offer a practical alternative, but since all such approaches rely on U-Net backbones, adapting them to DiTs is non-trivial, leaving a critical gap for transformer-based methods capable of true

end-to-end ultra-high-resolution generation.

**Position Extrapolation Schemes.** The challenge of ultra–high-resolution generation in DiTs closely mirrors that of *long-context generation* in language models, often tackled through advances in positional encoding. RoPE (Su et al., 2021) dominates this space, with extrapolation framed as frequency scaling: PI (Chen et al., 2023b) compresses positions to limit phase drift, while NTK-aware and YaRN (Peng et al., 2023) rescale frequencies to stabilize low modes and suppress unstable high ones. Inspired by these advances, vision models have begun to adopt these techniques. FiT (Lu et al., 2024) and FiT-v2 (Wang et al., 2024) introduce Vision-PI, Vision-NTK, and Vision-YaRN within DiTs by applying these frequency-scaling techniques independently to the horizontal and vertical axes. While this approach allows for flexible aspect-ratio generation and modest resolution gains, it remains a generic solution that overlooks the low-to-high frequency progression inherent to diffusion. RIFLEx (Zhao et al., 2025) demonstrates that frequency-aware extrapolation can also be effective in DiTs for video, enabling substantial temporal length extension. However, RIFLEx focuses exclusively on the temporal axis and does not address spatial resolution scaling. Lumina-Next (Zhuo et al., 2024) incorporates timestep dynamics by interpolating from PI to NTK-aware scaling as denoising advances. Yet, its heavy reliance on interpolation throughout the denoising process suppresses high frequencies, yielding blurry outputs. Our work, instead, directly analyzes the diffusion process frequency progression, leading to a principled approach that preserves fine-grained detail without compromising structural fidelity.

## 6. Conclusion

We presented DYPE, a training-free approach enabling diffusion transformers to synthesize ultra-high-resolution images without retraining or additional sampling overhead. Our method stems from a Fourier-space analysis of the samples' spectrum evolution during the diffusion sampling process, revealing that low-frequency content converges faster than the higher frequency bands. This regularity allows DYPE to better represent the evolving frequencies in the PE dynamically as well as enable the denoiser to operate more effectively within its training conditions.

As demonstrated on a pre-trained FLUX model, this strategy enables generation at unprecedented resolutions. Extensive qualitative and quantitative evaluations consistently confirm that DYPE offers superior generalization over existing static extrapolation techniques, with its advantage growing at higher resolutions.

As future work, we aim to pursue even more ambitious resolutions, not only through inference-time scaling but also by incorporating time-dependent positional extrapolation into a light tuning phase.

## Impact Statement

This paper presents work whose goal is to advance the efficiency and scalability of diffusion transformers for ultra-high-resolution image synthesis. Our research does not involve human subjects, personal data, or deployment in user-facing systems. All experiments were conducted on widely adopted, publicly available benchmarks.

We acknowledge that advancements in high-fidelity image generation carry potential societal consequences, particularly regarding the risk of misuse for generating deceptive or harmful content. However, our contributions are strictly focused on methodological improvements and evaluation on standardized datasets. We believe that improving the efficiency of these models does not inherently increase these risks beyond those already well-established in the field of generative modeling. We are committed to the principles of responsible AI research and transparency in methodology.

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

# A. Illustration of DYPE

Fig. 6 illustrates the behavior of RoPE frequencies under different scaling strategies, highlighting how our approach compares with position extrapolation methods.

# B. Comparison Between DYPE and Lumina-Next

The frequency allocation strategy behind DYPE is based on two complementary observations made in Sec. 3.1. The first related to the fact that low-frequency modes converge early in the sampling process, whereas the high frequency bands are resolved throughout the process. The second, is related to the trade-off exiting extrapolation method must take when trying to capture the entire spectrum of the larger resolution using the the fixed number of representable modes in the mode, $D$. Thus, rather than pinpointing both ends of the spectrum at all times and accommodating these trade-offs, DYPE, exploits the fact that low-frequencies are resolved earlier to better represent the higher bands and reduce the extrapolation compression.

The possibility of time-aware position extrapolation was briefly discussed in Zhuo et al. (2024) as part of introducing a new DiT architecture. However, the opposite conclusions were drawn by the authors. Specifically, their scheme starts by representing only the low-frequency band via PI (discarding high frequencies), and then switching to NTK-aware extrapolation that trades-off high frequency representation, in favor of low frequencies, which according to our analysis in Sec. 3.1 have already converged. We also note that in this scheme, the denoiser is not operating under the PE it was trained with unlike the case of DYPE.

Fig. 7 illustrates the complementary strategies of DYPE specifically DY-NTK-aware, and Time-Aware Scaled RoPE (Zhuo et al., 2024) in terms of the wavelengths they cover throughout the sampling.

**Quantitative and Qualitative Comparison with Time-Aware Scaled RoPE.** We conducted an experiment by applying DY-NTK-aware and Lumina-Next Time-Aware Scaled RoPE on top of the same pre-trained model, FLUX. Both methods are evaluated on the Aesthetic-4K benchmark using CLIPScore, ImageReward, Aesthetic-Score, and FID. For a better context, we also report the NTK-aware results.

The results in Tab. 5 show that DY-NTK-aware achieves the best performance across all metrics. Additionally, a qualitative comparison is provided in Fig. 8

# C. Implementation Details

Unless otherwise stated, all experiments are conducted on a single L40S GPU. We set $\alpha = 1$, $\beta = 32$, and use an effective resolution of $L = 1024$. Diffusion inference is performed with 28 sampling steps. For our method, we apply $\lambda_s = \lambda_t = 2$. Code will be released upon acceptance.

# D. Ablation Study

We perform an ablation study to better understand the role of specific design choices in DYPE, specifically, we consider alternative weighting schedulers for (i) DY-NTK-aware and (ii) DY-YaRN.

**Scheduler designs for DY-NTK-aware.** A central motivation for this ablation is to test how best to incorporate the low-to-high nature of diffusion into NTK-aware extrapolation. Recall from Sec. 2.2 that NTK-aware interpolation rescales each RoPE frequency $\theta_d$ as

$$h(\theta_d) = \frac{\theta_d}{s^{\,2d/(D-2)}}, \tag{18}$$

compressing low frequencies more while preserving higher ones. However, this scaling is fixed across all denoising steps and thus agnostic to the diffusion dynamics.

In DY-NTK-aware, we introduce a timestep-dependent scheduler $\kappa(t)$ to allow the effective frequency scaling to evolve with the diffusion timestep $t$. Here, we consider two ways the scheduler can interact with the NTK-aware rescaling factor $s$ from Eq. 6: (i) Multiplicative scaling, where the scheduler linearly modulates the compression,

$$h(\theta_d, t) = \frac{\theta_d}{(s \cdot \kappa(t))^{\frac{2d}{D-2}}}, \tag{19}$$

and (ii) Exponential scaling, where the scheduler exponentiates the compression,

$$h(\theta_d, t) = \frac{\theta_d}{s^{\,\kappa(t) \cdot \frac{2d}{D-2}}}. \tag{20}$$

In both cases, the scheduler is defined by the following family of time-parameterized scalings from Eq. 13:

$$\kappa(t) = \lambda_s \cdot t^{\lambda_t}, \tag{21}$$

with $\lambda_s$ and $\lambda_t$ controlling the magnitude and progression of the scheduler.

We ablate along two axes. First, we fix $\lambda_t = 1$ and vary $\lambda_s \in \{1, 1.5, 2, 2.5\}$ to identify the best magnitude scaling. Then, we fix $\lambda_s = 2$ (the winner) and vary $\lambda_t \in \{0.5, 1, 2\}$, corresponding to sublinear, linear, and exponential progression. We also compare against an NTK-aware variant with $\lambda_s = 2$ for completeness.

Results are summarized in Tab. 6, showing that increasing the initial scaling (toward position interpolation) improves

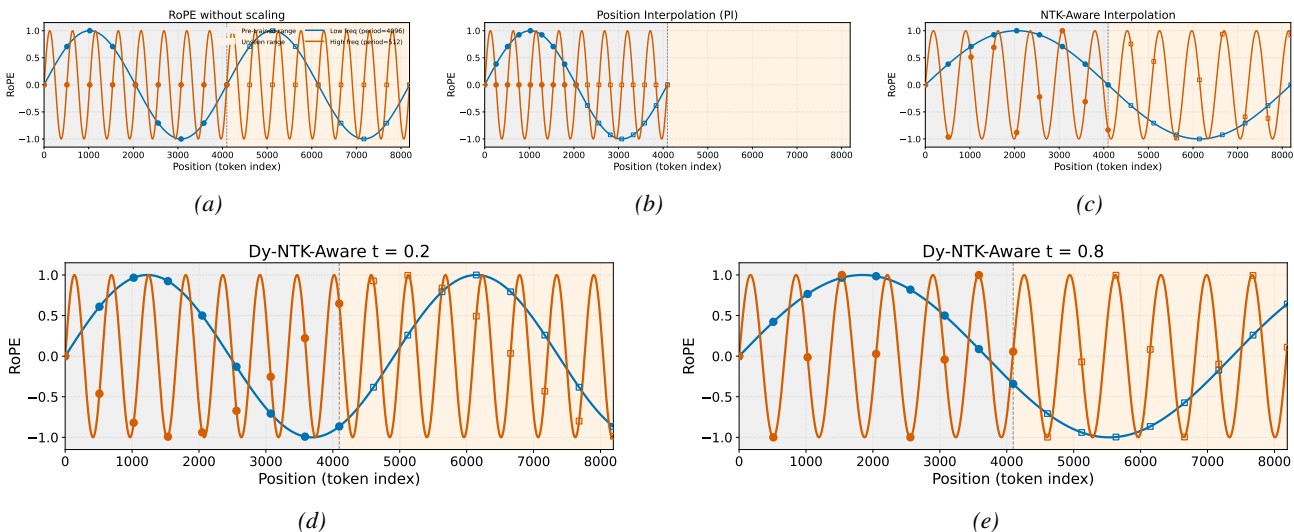

*Figure 6.* **Frequency Behavior Across Scaling Strategies.** **(a)** RoPE without scaling. **(b)** *Position Interpolation (PI)* where the sinusoidal curves are unchanged but the positions are normalized. **(c)** *NTK-Aware Interpolation* (frequency-dependent normalization; low frequency normalized more than high). **(d–e)** *Dy-NTK-Aware (ours)*: our method dynamically interpolates between RoPE and NTK-aware by blending their effective periods as a function of the diffusion timestep $t$ (shown here for $t=0.2$—close to image—and $t=0.8$—close to noise). Across panels, low frequency is shown in blue and high frequency in orange; training-context markers use filled circles, and test-context markers use hollow squares. Shaded backgrounds indicate pretrained (left) and unseen (right) position ranges.

*Table 5.* Comparison of NTK-aware, Time-Aware Scaled RoPE, and DY-NTK-aware on the Aesthetic-4K benchmark

| Method | CLIPScore ↑ | ImageReward ↑ | Aesthetic-Score ↑ | FID ↓ |
|---|---|---|---|---|
| NTK-aware | 24.88 | -0.54 | 5.50 | 203.85 |
| Time-Aware Scaled RoPE | 25.09 | -0.09 | 5.96 | 221.39 |
| DY-NTK-aware | **28.06** | **0.79** | **6.42** | **183.72** |

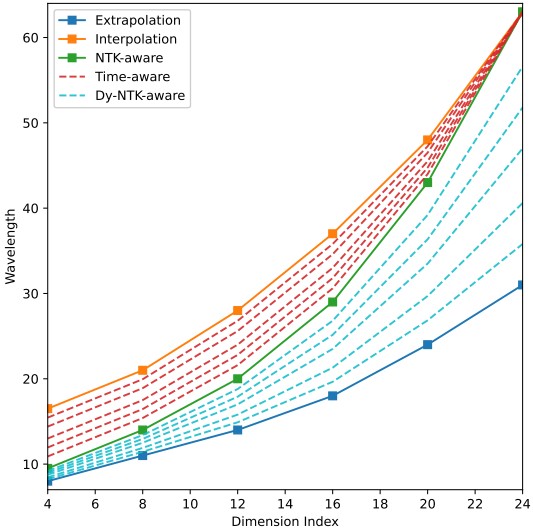

*Figure 7.* Wavelengths of the RoPE embeddings under different strategies. Solid curves show the baseline methods: Extrapolation (no scaling), PI, and NTK-aware. Dashed curves depict dynamic variants: *Time-aware* interpolates NTK-aware with PI, while *Dy-NTK-aware* interpolates NTK-aware with Extrapolation.

structural fidelity (CLIP), while faster attenuation with $t$ (toward complete position extrapolation) yields more aesthetic outputs. The exponential scheduler with $\lambda_s = 2$ and $\lambda_t = 2$ achieves the best balance between these objectives.

**Scheduler designs for DY-YaRN.** Building on the ablation study of DY-NTK-aware, we explore how to incorporate timestep dynamics into YaRN's frequency-dependent interpolation. Recall from Sec. 2.2 that YaRN introduces a weight $\gamma(r)$. YaRN smoothly interpolates between PI and no scaling. Specifically, YaRN rescales each RoPE frequency $\theta_d$ as:

$$h(\theta_d) = (1 - \gamma(r(d))) \frac{\theta_d}{s} + \gamma(r(d)) \theta_d, \quad (22)$$

where $r(d) = L/\lambda_d$. The ramp function $\gamma(r)$ smoothly transitions between PI-like stretching and no scaling:

$$\gamma(r) = \begin{cases} 0, & r < \alpha, \\ \frac{r-\alpha}{\beta-\alpha}, & \alpha \le r \le \beta, \\ 1, & r > \beta, \end{cases} \quad (23)$$

where $\alpha, \beta$ are hyperparameters setting the bands' boundaries.

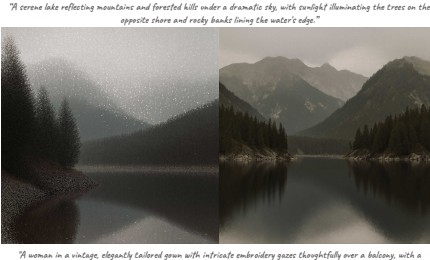

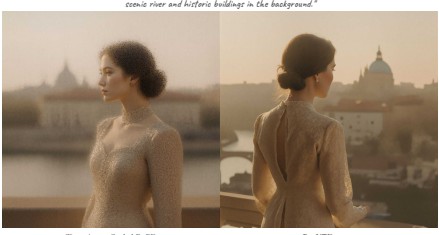

*Figure 8.* Qualitative comparison between DYPE and Time-Aware Scaled RoPE (Lumina-Next) on the Aesthetic-4K benchmark.

*Table 6.* Comparison of scheduler designs for DY-NTK-aware on FLUX at $3072^2$ resolution. Evaluated on 50 DrawBench prompts (sampled every 4th index). Baselines (FLUX, NTK-Aware) are included. Metrics: CLIP-Score (CLIP↑), ImageReward (IR↑), and Aesthetics-Score (Aesth↑).

| Variant | $\lambda_s$ | $\lambda_t$ | CLIP↑ | IR↑ | Aesth↑ |
|---|---|---|---|---|---|
| FLUX | - | - | 25.33 | -0.52 | 5.12 |
| NTK-Aware | - | - | 25.83 | -0.13 | 4.99 |
| Multiplicative | 1.0 | 1.0 | 25.67 | 0.11 | 5.08 |
| Multiplicative | 1.5 | 1.0 | 25.75 | 0.10 | 5.18 |
| Multiplicative | 2.0 | 1.0 | 26.09 | 0.16 | 5.31 |
| Multiplicative | 2.5 | 1.0 | 26.38 | 0.21 | 5.34 |
| Multiplicative | 2.0 | 0.5 | 26.28 | 0.21 | 5.34 |
| Multiplicative | 2.0 | 2.0 | 26.12 | 0.17 | 5.40 |
| Exponential | 1.0 | 1.0 | 25.81 | -0.13 | 5.03 |
| Exponential | 1.5 | 1.0 | 26.02 | 0.10 | 5.26 |
| Exponential | 2.0 | 1.0 | 26.52 | 0.29 | 5.39 |
| Exponential | 2.5 | 1.0 | 26.21 | 0.10 | 5.35 |
| Exponential | 2.0 | 0.5 | **26.69** | 0.24 | 5.34 |
| Exponential | 2.0 | 2.0 | 26.51 | **0.30** | **5.41** |

This can be viewed as partitioning frequencies into bands: low frequencies (small $d$) receive PI-like uniform scaling ($\gamma(r) = 0$), while high frequencies undergo no scaling ($\gamma(r) = 1$), while mid bands frequencies smoothly interpolate between the two by performing NTK-aware rescaling.

To leverage the low-to-high dynamics of diffusion, we introduce timestep dependence in three fashions: (i) Apply scheduler $\kappa(t)$ to the mid-level NTK-aware components, similarly to DY-NTK-aware in Eq. 15. (ii) Weight modulation: Apply scheduler $\kappa(t)$ to the ramp $\gamma$ parameters $\alpha, \beta$, effectively shifting the frequency bands assigned to each scaling regime as denoising progresses. (iii) Combined: Apply both $\kappa(t)$ to the threshold and NTK components simultaneously.

Following Sec. D, we use the best scheduler configuration (exponential with $\lambda_s = 2, \lambda_t = 2$). For the thresholds scheduler, we found that the best performing scheduler is, $\kappa(t) = t^2$. Intuitively, (i) controls how aggressively mid bands frequencies are compressed at each timestep, while (ii) controls which frequencies are considered "high", "mid" and "low" as a function of $t$.

Results in Tab. 7 show that considering only $\kappa(t)$ performs best. Further exhibiting our key idea—the fact that the diffusion process unfolds in a low-to-high manner, where early timesteps benefit from broader coverage of low frequencies, while later ones require sharper high-frequency detail. By modulating the ramp parameters $\alpha, \beta$ through $\kappa(t)$, the model adaptively reassigns frequencies between low, mid, and high bands in synchrony with the denoising trajectory. This dynamic partitioning allows YaRN to better capture large-scale structure early on while still allocating capacity to finer details as synthesis progresses, thereby yielding more coherent and visually appealing generations.

## E. Additional Results

**Evaluation on Additional DiT-Based Models.** To demonstrate the generalization capabilities of our approach beyond FLUX, we implemented DYPE on Qwen-Image (Wu et al., 2025). We compared the vanilla model, the static YaRN baseline, and our dynamic variant, DY-YaRN. All methods were evaluated on the Aesthetic-4K benchmark using the metrics established in our main experiments. As shown in Tab. 8, DY-YaRN achieves the best performance across all metrics (CLIPScore, ImageReward, Aesthetic-Score, and FID), validating the robustness and effectiveness of our method across different model architectures. Qualitative comparisons, illustrated in Fig. 9, further confirm that DY-YaRN produces superior visual quality with fewer artifacts compared to the baselines.

**High-Resolution Video Generation.** We extended our evaluation to the video domain by applying DYPE to the Wan2.1 1.3B model (Wan et al., 2025). While the original model has an effective resolution of $832 \times 480$, we assessed generation capabilities at a higher resolution of $1280 \times 720$. Due to GPU memory constraints, we fixed the sequence length to 33 frames. We compared our approach, specifically DY-YaRN, against the vanilla Wan2.1 model and YaRN using the VBench benchmark (Huang et al., 2023).

The quantitative results are summarized in Tab. 9. DY-YaRN outperforms the vanilla model across all categories. Notably, while YaRN suffers a degradation in the Imaging Quality score compared to the vanilla baseline, our dynamic approach improves it. Figure 10 provides a qualitative comparison.

*Table 7.* Comparison of scheduler application strategies for DY-YaRN on FLUX at $3072^2$ resolution. Evaluated on 50 DrawBench prompts (sampled every 4th index), with baselines (FLUX, YaRN) included. Metrics: CLIP-Score (CLIP↑), ImageReward (IR↑), and Aesthetics-Score (Aesth↑). All experiments use the best scheduler configuration from Sec. D (exponential with $\lambda_s = 2$, $\lambda_t = 2$).

| Variant | NTK term $\kappa(t)$ | By-parts $\kappa(t)$ | CLIP↑ | IR↑ | Aesth↑ |
|---|---|---|---|---|---|
| FLUX | - | - | 25.33 | -0.52 | 5.12 |
| YaRN | - | - | 27.32 | 0.36 | 5.47 |
| $\kappa(t)$ on NTK only | ✓ | - | 27.35 | 0.37 | 5.50 |
| $\kappa(t)$ on by-parts only | - | ✓ | **27.78** | **0.58** | **5.56** |
| $\kappa(t)$ on NTK & $\kappa(t)$ on by-parts | ✓ | ✓ | 27.76 | 0.36 | 5.41 |

*Table 8.* Comparison of Qwen-Image, YaRN, and DY-YaRN on the Aesthetic-4K benchmark.

| Method | CLIPScore ↑ | ImageReward ↑ | Aesthetic-Score ↑ | FID ↓ |
|---|---|---|---|---|
| Qwen-Image (Vanilla) | 27.98 | -0.26 | 5.52 | 201.15 |
| Qwen-Image + YaRN | 28.71 | 0.60 | 6.17 | 199.24 |
| Qwen-Image + DY-YaRN | **28.85** | **0.74** | **6.20** | **197.38** |

*Table 9.* Comparison of Wan2.1, YaRN, and DY-YaRN on high-resolution video generation ($1280 \times 720$) using VBench.

| Method | Subj. Const. ↑ | Bg. Const. ↑ | Mot. Smooth. ↑ | Dyn. Deg. ↑ | Aesth. Qual. ↑ | Img. Qual. ↑ |
|---|---|---|---|---|---|---|
| Wan 2.1 (Vanilla) | 0.9170 | 0.9518 | 0.9791 | 0.6532 | 0.4035 | 0.5107 |
| Wan 2.1 + YaRN | 0.9262 | 0.9513 | 0.9868 | 0.7350 | 0.4979 | 0.4364 |
| Wan 2.1 + DY-YaRN | **0.9303** | **0.9536** | **0.9899** | **0.8023** | **0.5095** | **0.6127** |

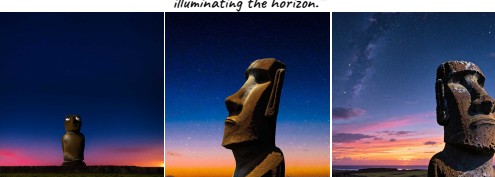

"A Moai statue gazes upward against a starry night sky, with a colorful sunset illuminating the horizon."

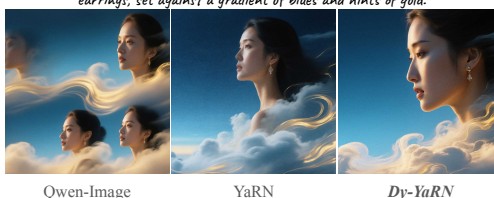

"A woman's profile emerges among swirling clouds, with a deep gaze and delicate earrings, set against a gradient of blues and hints of gold."

Qwen-Image     YaRN     *Dy-YaRN*

*Figure 9.* Qualitative comparison of Qwen-Image, YaRN, and DY-YaRN (both adapted on top of Qwen-Image) at a resolution of $4096 \times 4096$.

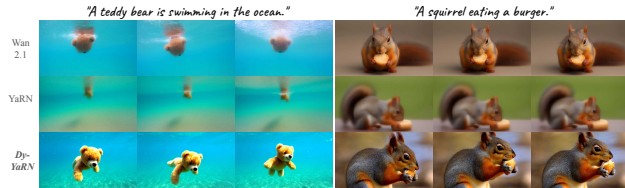

*Figure 10.* Qualitative comparison between Wan 2.1, YaRN, and DY-YaRN, both on top of Wan 2.1, applied to video generation at $1280 \times 720$.

**High-Resolution Image Editing.** To demonstrate DyPE's versatility, we further integrated it with the image editing model Qwen-Image-Edit-2509 (Wu et al., 2025). We evaluated performance on *multi-concept composition*, a challenging task requiring the seamless integration of distinct reference objects into a unified high-resolution scene. Experiments were conducted at an ultra-high resolution of $2656 \times 2656$. As shown in Fig. 11, the baseline (vanilla Qwen-Image-Edit-2509) tends to suffer from object duplication, such as the cats and baskets. In contrast, DyPE effectively mitigates these repetition artifacts, ensuring more precise object integration.

**Panoramic Image Generation.** We investigate DYPE's ability to handle extreme aspect ratios, focusing on panoramic images ($3 : 1$, $4096 \times 1365$). Such generation poses challenges for position encoding, as large horizontal spans can intensify aliasing and spatial inconsistencies. We evaluate on 20 prompts from Aesthetic-4K (every 10th entry), comparing DY-YaRN with YaRN and FLUX using CLIP-Score, ImageReward, and Aesthetics-Score. As shown in Table 10, DY-YaRN consistently outperforms YaRN, suggesting strong suitability for extreme spatial layouts. Figure 12 shows that YaRN fails to maintain correct aspect ratio proportion, leading to distorted object placement, while DY-YaRN preserves coherent spatial structure across the panorama.

**Additional Qualitative Results.** We present a collage of multi- and high-resolution outputs (see Fig. 13), all gener-

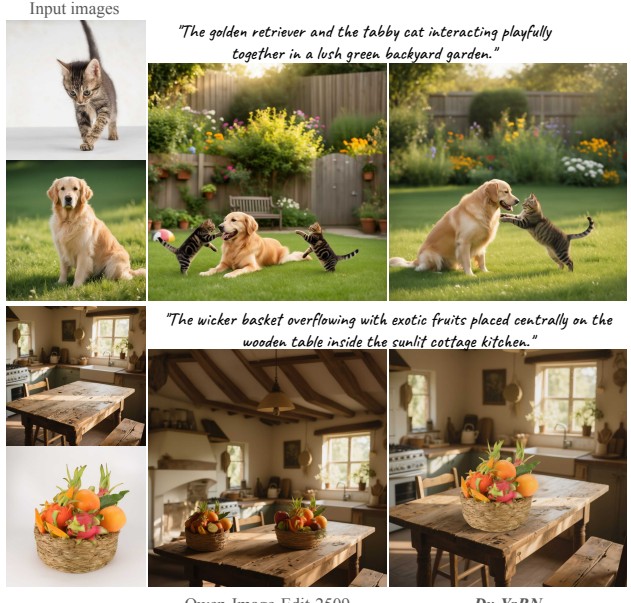

Input images

"The golden retriever and the tabby cat interacting playfully together in a lush green backyard garden."

"The wicker basket overflowing with exotic fruits placed centrally on the wooden table inside the sunlit cottage kitchen."

Qwen-Image-Edit-2509          *Dy-YaRN*

*Figure 11.* Qualitative comparison of high-resolution multi-concept composition at $2656 \times 2656$ between DY-YaRN and the *vanilla* Qwen-Image-Edit-2509.

*Table 10.* Panoramic image generation at $4096 \times 1365$ resolution.

| Method | CLIP-Score↑ | ImageReward↑ | Aesthetics-Score↑ |
|--------|-------------|--------------|-------------------|
| YaRN | 28.92 | 0.86 | 5.71 |
| DY-YaRN | **29.45** | **1.29** | **5.75** |

ated by DYPE.

**Qualitative Comparison with General Baselines.** Building on the comparisons presented in Sec. 4.1.2, we further provide qualitative results that highlight the differences between our approach and existing baselines (see Fig. 14).

**Qualitative results on the DrawBench benchmark.** Building upon the comparisons presented in Sec. 4.1, we provide further qualitative results comparing our approach to existing baselines.

**Additional Qualitative Results on the Aesthetic-4K Benchmark.** Expanding upon the comparisons discussed in Sec. 4.1, we present additional qualitative examples that highlight the performance of our method relative to existing baselines.

**Additional Zoom-in comparison.** Expanding upon the comparisons discussed in Fig. 17, we present additional qualitative examples that illustrate the differences if DY-YaRN with YaRN in fine details.

## F. Attention Entropy Analysis

Recent work (Jin et al., 2023) suggests that a primary reason trained attention mechanisms fail to generalize to higher resolutions is the shift in attention entropy relative to the training distribution. To investigate this, we analyze the *Normalized Attention Entropy* (scaled by the logarithm of the sequence length) as a function of the diffusion timestep, averaged across all layers and heads. We conducted this analysis using 20 random prompts from the Aesthetic-4K dataset, comparing the baseline FLUX model at its native resolution ($1024 \times 1024$) against FLUX, YaRN, and DY-YaRN at a resolution of $4096 \times 4096$.

As illustrated in Fig. 18, DY-YaRN best preserves the attention structure of the original distribution. Quantitatively, in terms of deviation from the baseline profile (measured by Mean Absolute Error), DY-YaRN achieves the lowest deviation (0.0455), outperforming both YaRN (0.0476) and the vanilla model (0.0529). This confirms that DYPE effectively mitigates the entropy shift typically observed during resolution extrapolation.

## G. Spectral Evolution for Different Schedulers

In the main text, our analysis of spectral progression (e.g., Figure 2) relies on the linear schedule commonly used in flow matching models ($\alpha_t = 1 - t$ and $\sigma_t = t$). Here, we examine this progression under alternative noise schedules, specifically the Variance Preserving (VP) and Variance Exploding (VE) formulations.

As established by Karras et al. (2022), VP and VE formulations are related via a simple scaling transform and therefore undergo the same fundamental spectral evolution. Because their signal-to-noise ratio (SNR) evolves non-linearly compared to the flow matching schedule, the specific curvature of the progression map $\gamma(f, t)$ bends differently across the timestep axis. Crucially, however, the monotonic ordering of the frequency evolution remains completely intact.

Figure 19 presents the empirical PSD progression maps for both VP and VE schedules. The visualizations confirm that the asymmetric convergence trend holds true across formulations: low-frequency modes converge much faster and saturate earlier, while high-frequency modes evolve gradually. This demonstrates that these frequency-dependent dynamics are not an artifact of a specific noise schedule, thoroughly validating DyPE's core premise across standard diffusion models.

*"A silhouetted pagoda stands against a large red moon, surrounded by dark mountains and trees, with some stylized birds flying in the sky. Stars twinkle in the night backdrop."*

YaRN

*Dy-YaRN*

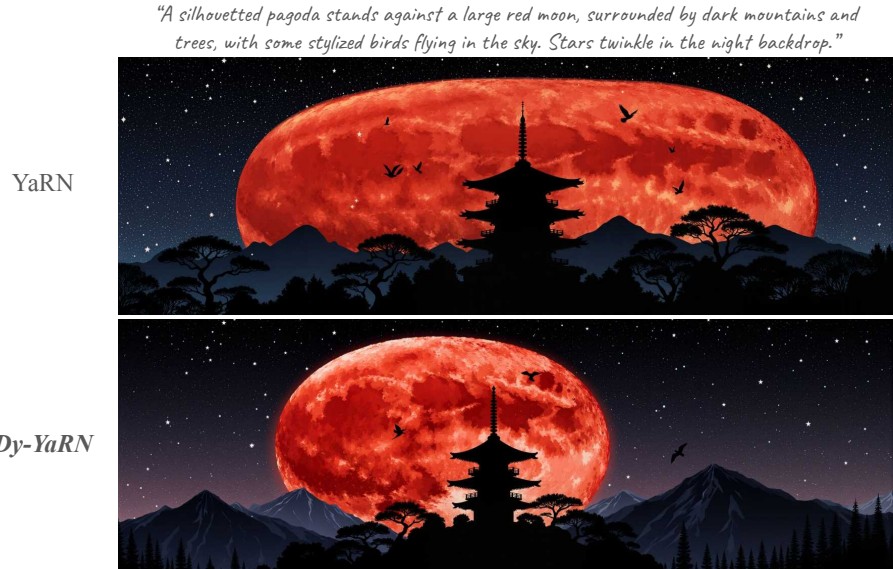

*Figure 12.* Qualitative comparison of panoramic generation at $4096 \times 1365$ resolution.

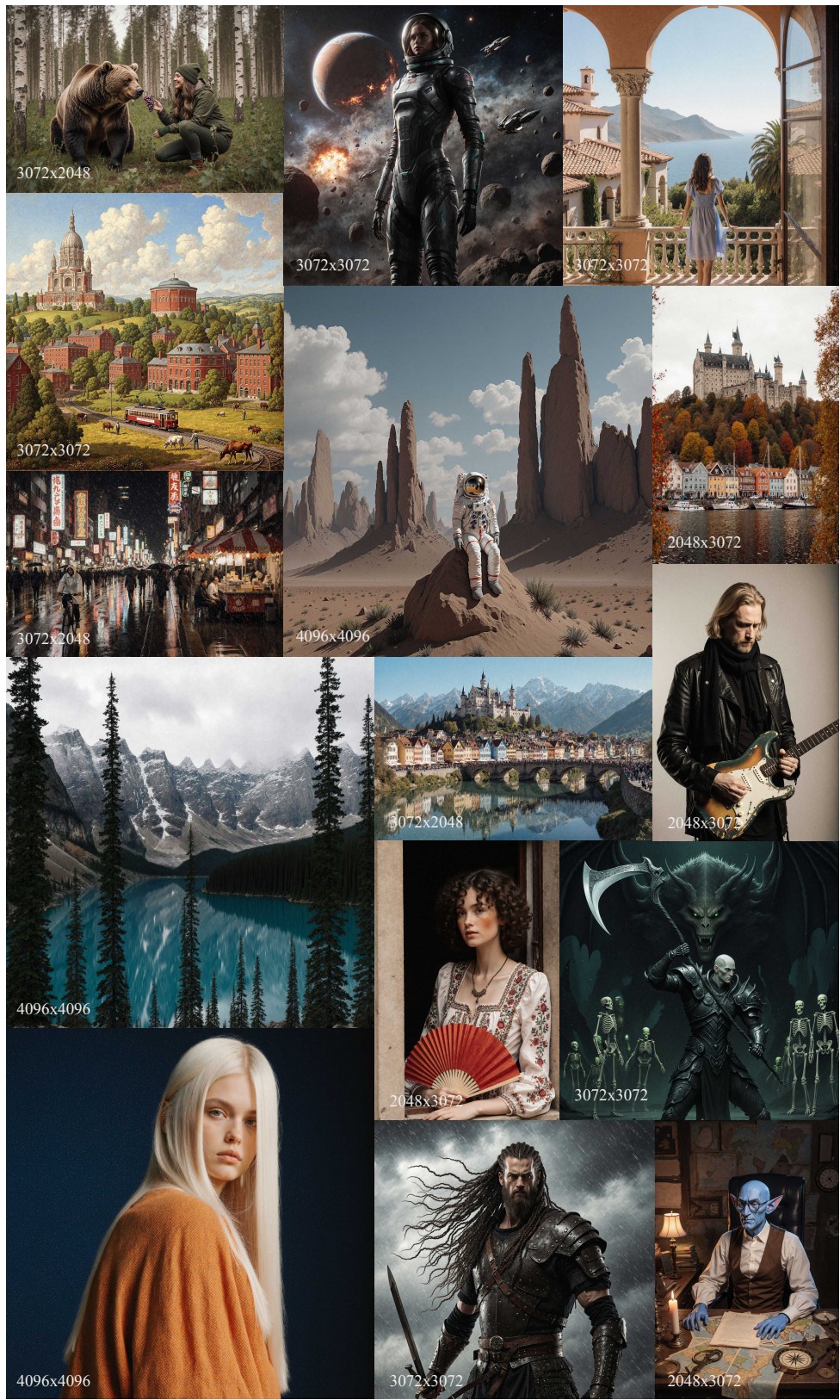

*Figure 13.* Collage of multi- and high-resolution results generated by DyPE. Prompts were taken from the Aesthetic-4K test set. Zoom-in for details.

*A vibrant, detailed landscape featuring a small town with red-brick buildings, green trees, and a rural backdrop. Prominently displayed in the background is a grand temple-like structure and a circular building, with a railroad track featuring a vintage streetcar running through the scene. Workers are seen in a field, and livestock grazes nearby, under a blue sky with fluffy clouds.*

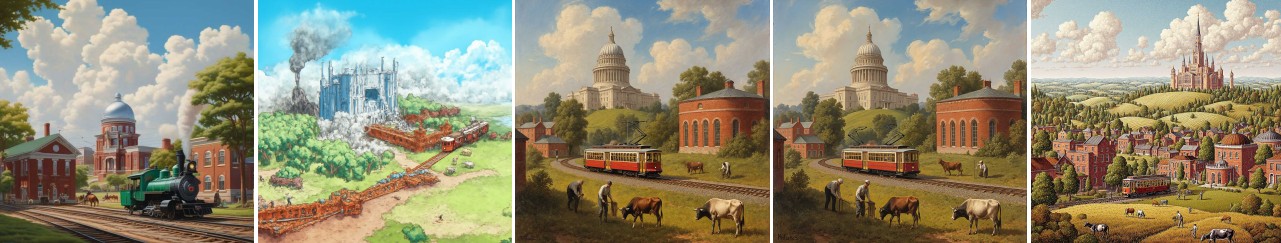

*Majestic mountains bathed in pink and purple hues under a starry night sky, with a glowing tower overlooking a serene waterfall and tranquil blue pool, surrounded by dark trees and rocky terrain.*

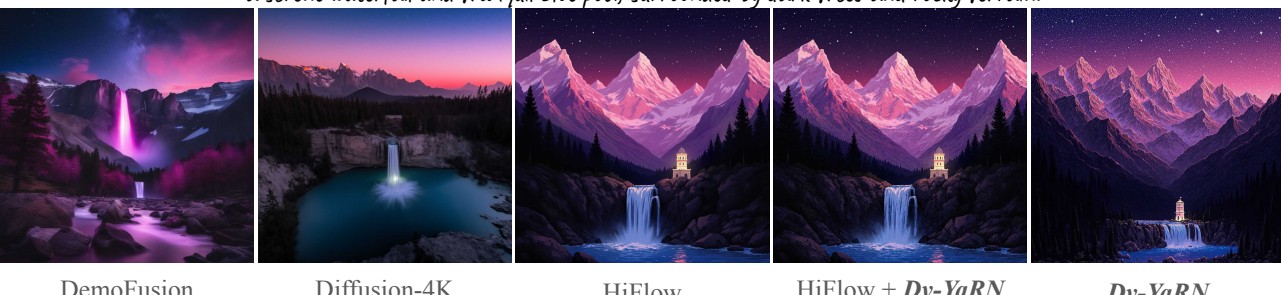

| DemoFusion | Diffusion-4K | HiFlow | HiFlow + *Dy-YaRN* | *Dy-YaRN* |

*Figure 14.* Qualitative results at $4096^2$ resolution using two representative prompts from Aesthetic-4K. We compare DemoFusion, Diffusion-4K, HiFlow, DY-YaRN+HiFlow and DY-YaRN.

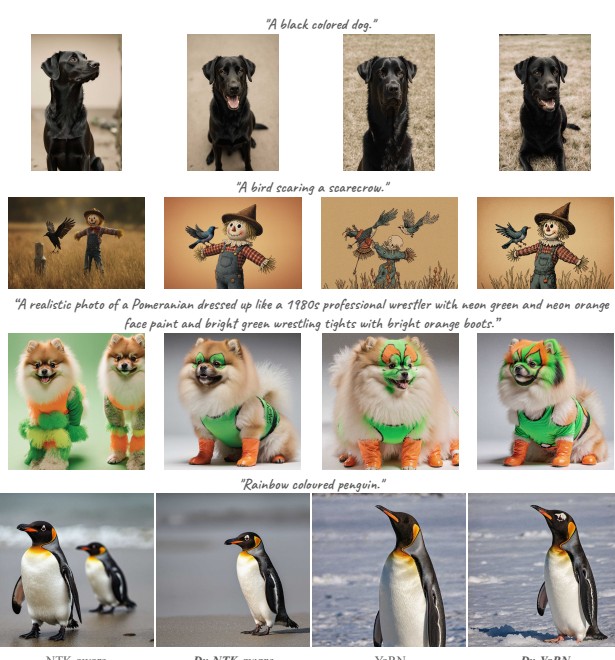

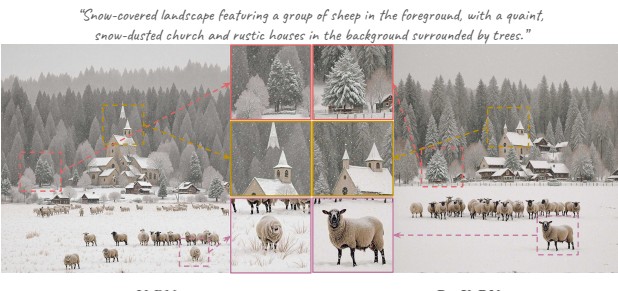

Figure 15. Qualitative results for high-resolution text-to-image generation on the DrawBench benchmark.

*Figure 17.* Zoom-in comparison at $4096^2$ resolution showing DY-YaRN vs. YaRN. Three magnified regions per image compare differences in fine details.

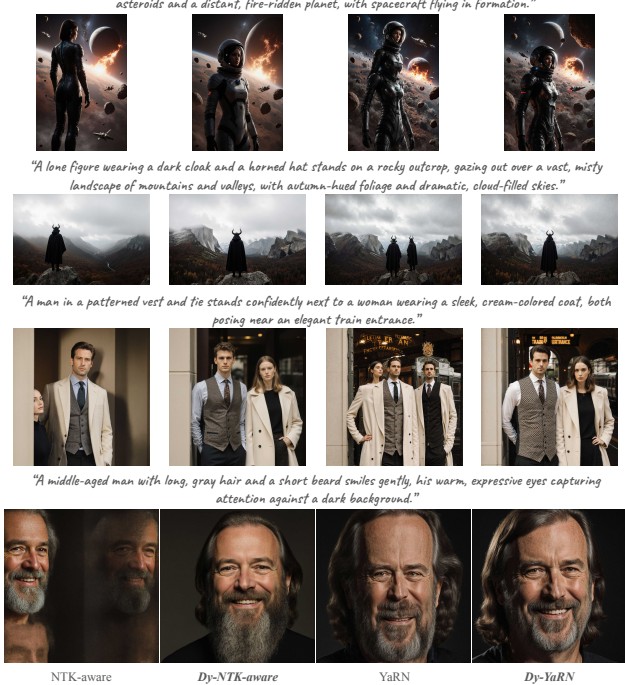

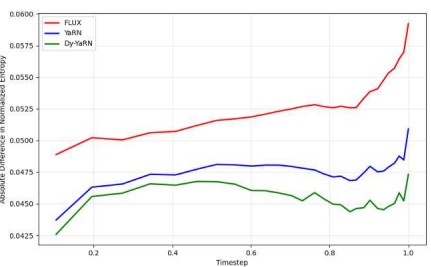

*Figure 18.* Deviation of Normalized Attention Entropy from the baseline ($1024 \times 1024$) profile across diffusion timesteps. Lower values indicate better preservation of the original attention structure.

*Figure 16.* High-resolution text-to-image generation results on the Aesthetic-4K benchmark.

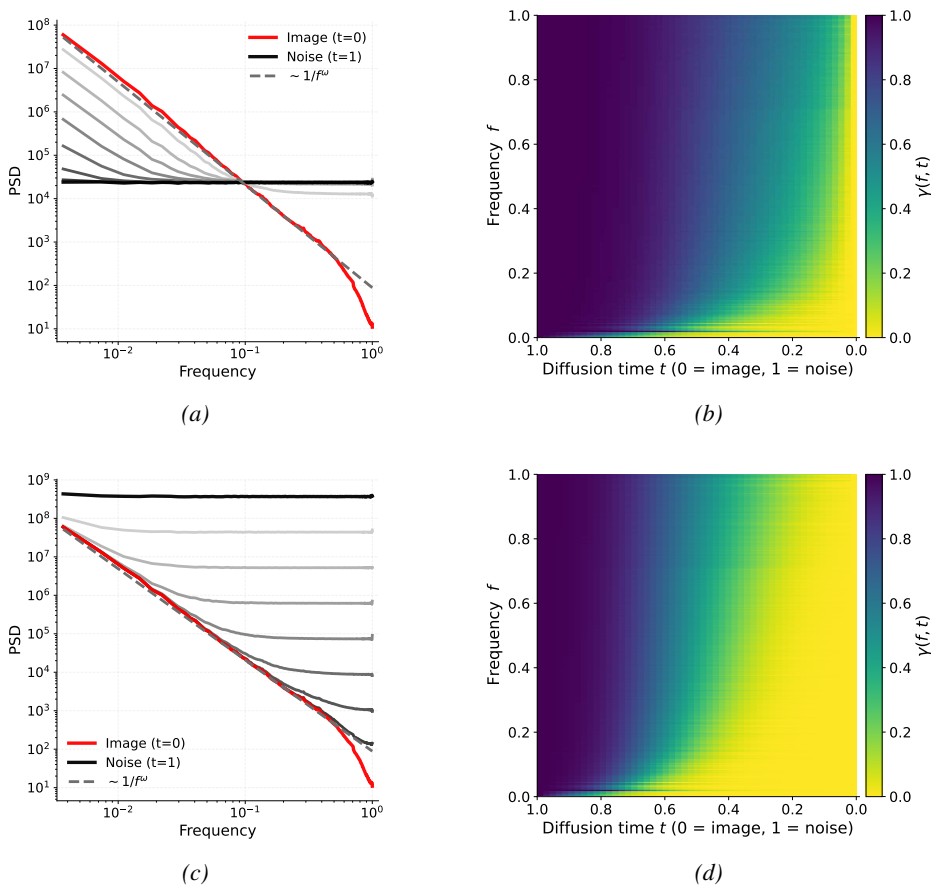

*Figure 19.* **Spectral Progression Maps for VP and VE Schedules.** Average PSD and progression maps $\gamma(f, t)$ illustrating the evolution of Fourier components for Variance Preserving **(a, b)** and Variance Exploding **(c, d)** noise schedules. Due to non-linear SNR evolution, the temporal curvature differs from the linear flow matching schedule, but the fundamental asymmetric convergence trend remains intact.

