# OpenReview forum: "DyPE: Dynamic Position Extrapolation for Ultra High Resolution Diffusion"
_ICML.cc/2026/Conference — ICML 2026 regular_

### Official Review · Reviewer_Gz4c · 2026-02-24

**Soundness:** 3
**Presentation:** 3
**Significance:** 3
**Originality:** 3
**Overall Recommendation:** 5
**Confidence:** 4

**Summary:**

The paper introduces DyPE, a training-free method designed to enable pre-trained DiTs to generate images at ultra-high resolutions beyond their training data limits.
The approach is based on the spectral progression of the diffusion process, noting that low-frequency structures converge early in sampling, whereas high-frequency details take more steps to resolve.
By dynamically adjusting the positional encoding such as RoPE at each diffusion step, DyPE matches the frequency spectrum to the current generative stage.
The method is applied to existing extrapolation techniques (PI, NTK-aware, and YaRN) by introducing a time-dependent scheduler, effectively applying stronger compression early on and transitioning to no scaling to preserve high frequencies later.
The authors demonstrate that DyPE successfully generates images exceeding 16 million pixels using the FLUX model without additional sampling costs and outperforms static baselines across multiple benchmarks.

**Compliance With Llm Reviewing Policy:**

Affirmed.

**Final Justification:**

The authors provided a comprehensive rebuttal that solidifies my positive assessment. I therefore maintain my original rating for acceptance.

**Key Questions For Authors:**

1. The mathematical motivation relies on the linear schedule of flow matching models where $\alpha_t=1-t$ and $\sigma_t=t$. How does the spectral progression of frequencies change under different diffusion noise schedules, such as Variance Preserving or Variance Exploding?
2. How robust is this method to the choice of sampling algorithm? Is the continuous formulation robust to different ODE/SDE solvers and varying numbers of inference steps?

**Limitations:**

Yes

**Strengths And Weaknesses:**

Soundness:
The authors provide a rigorous analysis of the power spectrum density of natural images during the reverse diffusion process, yielding a clear mathematical justification for why low frequencies converge faster.
However, the mathematical derivation relies explicitly on a linear schedule typical of flow matching models. The paper does not theoretically prove if this frequency convergence timeline behaves identically under other schedules and diffusion models.

Presentation:
The paper is logically structured and clearly written. The inclusion of the PSD map to visualize spectral evolution serves as a fantastic pedagogical tool to justify the method's design.


Significance:
The paper propose a training-free, effective solution that scales DiTs to 16 million pixels is highly significant for the community. The empirical results are comprehensive.

Originality:
The application of time-dependent dynamic scaling to positional extrapolation is a highly original angle. DyPE's formulation is a novel and more accurate reflection of time-aware position extrapolation.

---

> ### Author Rebuttal · Authors · 2026-03-31
>
> We thank the reviewer for recognizing the depth of our analysis and the significance of our approach.
>
> **Q1: Spectral Progression under VP and VE Schedules**
> While we utilized the linear schedule of flow matching for our formal derivation due to its simplicity and broad usability, the core low-to-high frequency progression is a universal property of diffusion models, not a byproduct of the linear schedule. In all continuous-time diffusion formulations, the forward noise-injection process acts as a progressive frequency degrader, dictating that the reverse process must inherently resolve low frequencies before high frequencies.
>
> Furthermore, as established in EDM (Karras et al., 2022), Variance Preserving (VP) and Variance Exploding (VE) formulations are related to one another by a simple scaling transform and therefore undergo the same fundamental spectral evolution. Under these schedules, the signal-to-noise ratio (SNR) evolves non-linearly compared to Flow Matching. This means the specific curvature of the progression map $\gamma(f,t)$ will bend differently across the timestep axis $t$. However, the monotonic ordering remains completely intact.
>
> To confirm this empirically, we have generated the exact PSD progression maps for both VP and VE schedules, which we have included in the provided doc (https://docs.google.com/document/d/e/2PACX-1vRBxJwK3qcRoR816pEXtiw-38_CPlTOoUiTSAqHHKkSOtQCJStu04w_YMaRHSTLcNQKRH7M8PLD2B0b/pub). As expected, the visualizations confirm that the asymmetric convergence trend holds, validating DyPE’s core premise across diffusion formulations. We will include this extended analysis in the camera-ready version.
>
> **Q2: Robustness to Sampling Algorithms**
> Fundamentally, DyPE introduces an architectural modification to the positional encoding layer, but the network itself remains a standard diffusion model. Because it continues to act as a standard diffuser, it can be used seamlessly out-of-the-box with any sampling algorithm. Specifically, because DyPE relies strictly on the current normalized timestep $t \in [0, 1]$ for frequency scaling, its operation is entirely decoupled from how the underlying ODE/SDE solver navigates the generative timeline.
>
> To validate this empirically, we evaluated our Dy-YaRN variant against vanilla FLUX and static YaRN across two radically different solvers to complement the baseline Euler results: an SDE solver (Euler Stochastic) and a higher-order ODE solver (Heun).
>
> Evaluated on 20 Aesthetic-4K prompts at 4096×4096 resolution:
>
> **Euler SDE (1st order, stochastic, 50 steps):**
>
> | Method         | CLIP Score ↑ | ImageReward ↑ | Aesthetic Score ↑ |
> |----------------|-------------:|--------------:|------------------:|
> | FLUX  | 23.84        | -1.36         | 4.88              |
> | YaRN  | 27.57        | -0.45         | 5.86              |
> | **Dy-YaRN (Ours)** | **27.68** | **-0.35** | **5.89** |
>
> **Heun ODE (2nd order, deterministic, 28 steps):**
>
> | Method         | CLIP Score ↑ | ImageReward ↑ | Aesthetic Score ↑ |
> |----------------|-------------:|--------------:|------------------:|
> | FLUX | 26.28        | -0.63         | 5.52              |
> | YaRN  | 29.10        | 0.68          | 6.47              |
> | **Dy-YaRN (Ours)** | **29.20** | **0.81** | **6.72** |
>
> As demonstrated, DyPE consistently outperforms the static baselines regardless of the solver type. We will include these comparisons in the camera-ready version.
>
> We remain available to address any further questions during the discussion period.

---

> > ### Author Rebuttal · Reviewer_Gz4c · 2026-04-03
> >
> > I appreciate the detailed rebuttal and the additional experimental results provided. I decide to maintain my original rating.

---

### Official Review · Reviewer_Uo4p · 2026-03-10

**Soundness:** 3
**Presentation:** 4
**Significance:** 3
**Originality:** 3
**Overall Recommendation:** 4
**Confidence:** 4

**Summary:**

This paper investigates how to effectively extrapolate positional encodings in Diffusion Transformers (DiTs) to generate ultra-high-resolution images without additional training. To address this, the authors propose DyPE (Dynamic Position Extrapolation), a training-free method. Based on a frequency-domain analysis of the diffusion reverse sampling process, the authors observe that low-frequency structures converge early in denoising, while high-frequency details evolve continuously. Building on this, DyPE improves existing static position extrapolation methods (e.g., NTK-aware and YaRN) by making them timestep-dependent. In the early denoising stages, DyPE applies strong positional scaling to ensure global structural coherence; as denoising progresses, it gradually decays the scaling factor to generate rich details in the original high-frequency feature space.

**Compliance With Llm Reviewing Policy:**

Affirmed.

**Final Justification:**

The rebuttal addresses the main practical questions reasonably.

**Key Questions For Authors:**

Q1. Heuristic nature of the scheduler design:
In Eq. (13), a scheduler is introduced to control the scaling ratio. However, this design appears highly heuristic and lacks rigorous theoretical derivation. Have the authors considered deriving this scheduler? Are there ablation studies justifying this specific non-linear decay strategy over other potential alternatives?

Q2. Generalization to non-RoPE encodings:
DyPE heavily relies on the frequency characteristics of RoPE. For DiT models using standard Absolute Sinusoidal Positional Encodings (like the original ViT) or Learnable PEs, can the core philosophy of DyPE (strong scaling for global structure early on, decaying for local details later) be generalized? If so, how would it be implemented?

Q3. Practical usability vs. Lightweight fine-tuning (The Upper Bound):
The authors emphasize that the training-free nature avoids prohibitive computational costs. However, in practice, high-resolution generation can often be achieved efficiently via lightweight fine-tuning (e.g., training a LoRA with a few high-res images or brief resolution warm-up) with minimal resource consumption. How large is the performance gap (e.g., FID, detail realism) between DyPE and such lightweight fine-tuning methods? Could the authors provide a "lightly fine-tuned model" as an upper bound to demonstrate whether the trade-off between zero training cost and  generation quality is acceptable?

**Limitations:**

The authors do not include a dedicated Limitations section in the main text, only briefly mentioning dataset bias (portraits vs. landscapes) in the Impact Statement in the appendix. I recommend adding a dedicated discussion in the final version addressing:

(1) What are the specific failure modes or model collapse behaviors when the resolution is extrapolated to extreme, unprecedented scales?
(2) Where is the performance ceiling of this method, and is the gap between DyPE and lightweight/full fine-tuning methods acceptable in real-world deployments?

**Strengths And Weaknesses:**

1. The motivation is well-founded. The quantitative analysis of the spectral evolution during the diffusion process clearly reveals the convergence differences between low-frequency and high-frequency components across timesteps, providing a solid basis for the proposed method.
2. The paper is well-written with clear logic, and the visual charts/figures are of high quality.
3. The proposed DyPE method demonstrates practical utility and offers a cost-effective solution for high-resolution generation tasks.

Weaknesses:
1.	Incremental Novelty of the Observation: The core observation (diffusion models generate from low to high frequencies) is a relatively well-known property in the diffusion community. While applying this to dynamic RoPE truncation is effective, the theoretical novelty is somewhat incremental.
2.	Lack of an Upper Bound / Practical Baseline: The paper restricts its scope entirely to "training-free" methods and lacks a comparison with widely used "lightweight fine-tuning" approaches (e.g., High-res LoRA), making it difficult to assess the true practical value and performance ceiling of the proposed method.

---

> ### Author Rebuttal · Authors · 2026-03-31
>
> We thank the reviewer for recognizing the solid foundation and practical utility of our work.
>
> **W1: Novelty of the Observation** We respectfully address the novelty from two complementary perspectives. First, from a practical standpoint, we show how the general intuition of low-to-high frequency generation can be translated into a dynamic capacity-allocation strategy for DiT positional encodings. Second, from a theoretical standpoint, our novel observation goes beyond the well-known spectral progression. As shown in Figure 2b, our novel finding is the strict asymmetry of this convergence: high-frequency bands continue to evolve throughout the entire denoising process, whereas low frequencies settle and cease to evolve very early. This specific insight is exactly what dictates the design of DyPE, allowing the method to dynamically reallocate positional focus to resolving high frequencies without sacrificing global structural coherence.
>
> **W2 & Q3: Lightweight Fine-Tuning as an Upper Bound** The reviewer asks for a comparison against a fine-tuning approach to establish a performance ceiling. We point the reviewer to Table 3, where we evaluate *Diffusion-4K*. Diffusion-4K is a fine-tuning approach (incorporating LoRA and resolution warm-up) specifically trained on the Aesthetic-4K dataset. Remarkably, our training-free DyPE remains superior to this fine-tuned baseline across multiple metrics at both 2048² and 4096² resolutions.
>
> **Q1: Heuristic Nature of the Scheduler** First and foremost, the schedule $\kappa$ presented in our paper qualitatively implements the spectral adaptation in the PE, reflecting the unique frequency dynamics of the diffusion model. The exact analytical relationship between the diffusion evolution and the optimal PE representation is difficult to pinpoint, as it heavily depends on the specific application, target metrics, and other factors. For this reason, we parameterized this relationship using a two-parameter family of functions - an approach that proved highly effective in enabling the strong performance achieved by our method. Furthermore, Tables 6 and 7 in the Appendix provide extensive ablations justifying our specific non-linear decay strategy over sublinear or uniform alternatives.
>
> **Q2: Generalization to non-RoPE encodings**
> DyPE is primarily derived for for RoPE-based architectures, the de-facto positional encoding in modern diffusion transformers (FLUX, QWEN, Lumina, FiT), similarly to how NTK and YaRN were developed for RoPE in the LLM domain. RoPE injects positional information multiplicatively at every attention layer, allowing direct frequency-level modulation. In contrast, absolute sinusoidal PEs are static, precomputed embeddings added only once at the input layer, severely limiting post-hoc frequency modulation. Therefore, DyPE's frequency scaling cannot be directly applied to such encodings.
>
> However, the core philosophy of DyPE, aligning positional scaling with the spectral progression of the diffusion process, is generalizable through a PI-style adaptation. To validate this, we adapted our approach for Stable Diffusion 3.5 Large (SD3.5 Large), a DiT which utilizes 2D absolute sinusoidal PEs. Rather than compressing position indices (classical PI, which also produces out-of-distribution sincos values due to nonlinearity), our adaptation operates entirely within the model's precomputed positional embedding buffer. At each denoising timestep, we blend two crops from this buffer: a bilinearly-upsampled training-resolution crop and a natural higher-resolution crop, weighted by $\kappa(t) = t^2$. This is conceptually our Dy-PI realized in embedding space.
>
> We evaluated on 20 Aesthetic-4K prompts at 1536×1536 (1.5× the 1024 training resolution):
>
> | Method                | CLIP Score ↑ | ImageReward ↑ | Aesthetic Score ↑ |
> |-----------------------|-------------:|--------------:|------------------:|
> | Vanilla | 23.42        | -0.83         | 4.66              |
> | Static Interpolation  | 29.01        | 0.62          | 6.11              |
> | **DyPE-Adapted (Ours)**| **29.08** | **0.75** | **6.25** |
>
> DyPE consistently outperforms both baselines, confirming that the underlying philosophy of DyPE successfully generalizes beyond RoPE.
>
> **Limitations and Failure Modes** We will add a dedicated Limitations section in the main text. Regarding extreme scales, we point the reviewer to our scaling analysis in Figure 5. As the resolution scales to unprecedented extremes (e.g., 6144²), the primary failure mode is a gradual degradation in ImageReward, eventually leading to structural incoherence and semantic breakdown, rather than the immediate grid-like tiling seen in baseline models.
>
> We remain available to address any further questions during the discussion period.

---

> > ### Author Rebuttal · Reviewer_Uo4p · 2026-04-04
> >
> > The rebuttal addresses the main practical questions reasonably. Table 3 already provides a relevant fine-tuned upper-bound comparison (Diffusion-4K), and the appendix ablations in Tables 6–7 support the chosen scheduler more clearly than the original draft. The SD3.5 adaptation also strengthens the claim that the core idea is not strictly RoPE-specific.
> >
> > These clarifications do not materially change my central view that the conceptual novelty is still somewhat incremental and that several key validations appear only in rebuttal or are deferred to the possible camera-ready version.
> >
> > I remain positive because the method is simple, training-free, broadly useful, and empirically strong at extreme resolutions, but I do not see a case for upgrading beyond weak accept.

---

### Official Review · Reviewer_zTiH · 2026-03-13

**Soundness:** 3
**Presentation:** 4
**Significance:** 4
**Originality:** 4
**Overall Recommendation:** 5
**Confidence:** 3

**Summary:**

The paper proposes DyPE, a method to adapt Rotary Positional Embeddings in DiT dynamically during the denoising process. By recognizing that different image frequencies settle at different timesteps, the authors vary the PE scaling factor over time. This allows the model to maintain structural coherence early in sampling while avoiding the blurriness typically associated with static interpolation during the final refinement steps.

**Compliance With Llm Reviewing Policy:**

Affirmed.

**Key Questions For Authors:**

1. Does DyPE offer any strategy to integrate with memory-efficient attention such as FlashAttention or SageAttention?
2. Is per-model tuning required regarding the two hyperparameters of the scheduler?
3. As shown in Figure 1, FLUX can sometimes stuck with repetitions. Does DyPE purely solve the PE-induced repetition, or does the model still struggle with semantic  repetition?

**Limitations:**

yes

**Strengths And Weaknesses:**

Strengths:
- This paper presents a technically sound and highly practical contribution to the field of high-resolution image synthesis. The clarity of the paper is high, and the qualitative results are compelling.
- The core idea of coupling PE extrapolation to the spectral progression of diffusion is elegant and well-supported by Fourier analysis.
- The authors demonstrate DyPE on a variety of state-of-the-art DiTs and even extend it to video and image editing, showing the method's versatility.
- DyPE is training-free and introduces zero additional sampling cost, making it a superior alternative to refinement-based or tuning-based high-resolution methods.

Weaknesses:
- Despite significant improvements demonstrated in empirical validation across multiple models, the novelty is incremental (extending YaRN/NTK with a temporal scheduler).
- The scheduler introduces two hyperparameters, which may require model-dependent tuning, potentially limiting its practicality.
- While the model can mathematically handle the large amount of tokens, the paper does not address the quadratic memory scaling of self-attention.

---

> ### Author Rebuttal · Authors · 2026-03-31
>
> We thank the reviewer for their thoughtful review.
>
> **Integration with Memory-Efficient Attention:** DyPE is compatible with any memory-efficient attention implementations, such as FlashAttention and SageAttention. Our method dynamically scales the RoPE frequencies as a function of the denoising timestep, but does not intervene with the attention mechanism. Therefore, our approach is compatible with memory-efficient attention mechanisms. Additionally, as shown in Table 3, DyPE is compatible with more efficient inference pipelines like HiFlow, further reducing compute requirements.
>
> **Hyperparameter Tuning:** Extensive per-model tuning is not required. First, we provide an extensive ablation study in Table 6 that assesses the effect of the hyperparameters on FLUX. Crucially, we utilized the exact same optimal configuration ($\lambda_s=2, \lambda_t=2$) across all other models evaluated in the paper, including FiTv2, Qwen-Image, and Wan, as well as across diverse tasks such as class-conditional generation, image editing, and video generation. This single configuration consistently outperformed static baselines across different architectures and modalities. We will clarify this generalization in the camera-ready version.
>
> **PE-Induced vs. Semantic Repetition:** DyPE specifically resolves PE-induced spatial repetition. As recently observed in RIFLEx (Zhao et al., 2025), repetition during extrapolation occurs because generating beyond the intrinsic frequency of the positional embeddings causes the content to naturally loop. As illustrated in the baseline FLUX generation in Figure 1, static extrapolation causes this exact periodicity to manifest as grid-like, duplicated artifacts at ultra-high resolutions. Because DyPE dynamically interpolates the effective periods of the embeddings (as discussed and illustrated in Appendix A), it breaks this periodicity and allows the model to maintain structural coherence across large spatial extents, avoiding the grid-like tiling artifacts. Indeed, across all our experiments, we did not encounter any repetition artifacts when using DyPE.
>
> We remain available to address any further questions during the discussion period.

---

> > ### Author Rebuttal · Reviewer_zTiH · 2026-04-04
> >
> > The rebuttal effectively resolves the primary technical concerns regarding the scalability and generalizability of the method. While the conceptual novelty is incremental, the method provides a strong, training-free solution that effectively mitigates grid-like artifacts and maintains structural coherence at ultra-high resolutions.

---

### Decision · Program_Chairs · 2026-04-30

**Decision:**

Accept (regular)

**Comment:**

The main strength is a simple, training-free, zero-overhead method that significantly improves ultra-high-resolution generation by making positional extrapolation timestep-aware, grounded in a convincing spectral analysis of diffusion dynamics. Reviewers consistently view the paper as technically solid, clearly written, and practically impactful. The empirical study is broad and compelling across multiple DiTs, with extensions to video and editing further supporting significance. Concerns focus mainly on incremental novelty, heuristic scheduler design, lack of comparison to lightweight fine-tuning upper bounds, and limited discussion of memory scaling and generalization across models/schedules. Overall, the contribution is strong, useful, and likely to influence follow-up work.

The authors should check and refine the references very carefully. If not resolved, it has a risk of desk rejection. There are hallucinated/problematic reference(s):
Reference: Chen, S., Lin, Z., Chen, Z., Ren, S., He, J., Chen, Z., Ma, S., Chen, W., Tang, J., and Sun, M. Extending context window of large language models via positional interpolation. arXiv preprint arXiv:2306.15595, 2023b.
Issue: authors mismatch with arXiv

"Scaling Rectified Flow Transformers for High-Resolution Image Synthesis" appears twice in the reference part.